# A Hypergradient Approach to Robust Regression without Correspondence

**Yujia Xie**,[*] **Yixiu Mao**,[*] **Simiao Zuo, Hongteng Xu, Xiaojing Ye, Tuo Zhao, Hongyuan Zha**[†]

## Abstract

We consider a regression problem, where the correspondence between input and output data is not available. Such shuffled data is commonly observed in many real world problems. Taking flow cytometry as an example, the measuring instruments are unable to preserve the correspondence between the samples and the measurements. Due to the combinatorial nature, most of existing methods are only applicable when the sample size is small, and limited to linear regression models. To overcome such bottlenecks, we propose a new computational framework – ROBOT– for the shuffled regression problem, which is applicable to large data and complex models. Specifically, we propose to formulate the regression without correspondence as a continuous optimization problem. Then by exploiting the interaction between the regression model and the data correspondence, we propose to develop a hypergradient approach based on differentiable programming techniques. Such a hypergradient approach essentially views the data correspondence as an operator of the regression, and therefore allows us to find a better descent direction for the model parameter by differentiating through the data correspondence. ROBOT is quite general, and can be further extended to the inexact correspondence setting, where the input and output data are not necessarily exactly aligned. Thorough numerical experiments show that ROBOT achieves better performance than existing methods in both linear and nonlinear regression tasks, including real-world applications such as flow cytometry and multi-object tracking.

## 1 Introduction

Regression analysis has been widely used in various machine learning applications to infer the the relationship between an explanatory random variable (i.e., the input) $X \in \mathbb{R}^d$ and a response random variable (i.e., the output) $Y \in \mathbb{R}^o$ (Stanton, 2001). In the classical setting, regression is used on labeled datasets that contain paired samples $\{x_i, y_i\}_{i=1}^n$, where $x_i$, $y_i$ are realizations of $X$, $Y$, respectively.

Unfortunately, such an input-output correspondence is not always available in some applications. One example is flow cytometry, which is a physical experiment for measuring properties of cells, e.g., affinity to a particular target (Abid & Zou, 2018). Through this process, cells are suspended in a fluid and injected into the flow cytometer, where measurements are taken using the scattering of a laser. However, the instruments are unable to differentiate the cells passing through the laser, such that the correspondence between the cell proprieties (i.e., the measurements) and the cells is unknown. This prevents us from analyzing the relationship between the instruments and the measurements using classical regression analysis, due to the missing correspondence. Another example is multi-object tracking, where we need to infer the motion of objects given consecutive frames in

---

[*]Equal Contributions.

[†]Yujia Xie, Simiao Zuo, Tuo Zhao are affiliated with Georgia Institute of Technology. Emails: {xieyujia, simiaozuo, tourzhao}@gatech.edu. Yixiu Mao is affiliated with Shanghai Jiao Tong University. Email: 956986044myx@gmail.com. Hongteng Xu is affiliated with Gaoling School of Artificial Intelligence, Renmin University of China, and Beijing Key Laboratory of of Big Data Management and Analysis Methods. Email: hongtengxu@ruc.edu.cn. Xiaojing Ye is affiliated with Georgia State University. Email: xye@gsu.edu. Hongyuan Zha is affiliated with School of Data Science, Shenzhen Institute of Artificial Intelligence and Robotics for Society, the Chinese University of Hong Kong, Shenzhen. Email: zhahy@cuhk.edu.cn.

a video. This requires us to find the correspondence between the objects in the current frame and those in the next frame.

The two examples above can be formulated as a shuffled regression problem. Specifically, we consider a multivariate regression model

$$Y = f(X, Z; w) + \varepsilon,$$

where $X \in \mathbb{R}^d$, $Z \in \mathbb{R}^e$ are two input vectors, $Y \in \mathbb{R}^o$ is an output vector, $f : \mathbb{R}^{d+e} \to \mathbb{R}^o$ is the unknown regression model with parameters $w$ and $\varepsilon$ is the random noise independent on $X$ and $Z$. When we sample realizations from such a regression model, the correspondence between $(X, Y)$ and $Z$ is not available. Accordingly, we collect two datasets $\mathcal{D}_1 = \{x_i, y_i\}_{i=1}^n$ and $\mathcal{D}_2 = \{z_j\}_{j=1}^n$, and there exists a permutation $\pi^*$ such that $(x_i, z_{\pi(i)})$ corresponds to $y_i$ in the regression model. Our goal is to recover the unknown model parameter $w$. Existing literature also refer to the shuffled regression problem as *unlabeled sensing*, *homomorphic sensing*, and *regression with an unknown permutation* (Unnikrishnan et al., 2018). Throughout the rest of the paper, we refer to it as *Regression WithOut Correspondence* (RWOC).

A natural choice of the objective for RWOC is to minimize the sum of squared residuals with respect to the regression model parameter $w$ up to the permutation $\pi(\cdot)$ over the training data, i.e.,

$$\min_{w,\pi} \mathcal{L}(w, \pi) = \sum_{i=1}^n \|y_i - f(x_i, z_{\pi(i)}; w)\|_2^2. \tag{1}$$

Existing works on RWOC mostly focus on theoretical properties of the global optima to equation 1 for estimating $w$ and $\pi$ (Pananjady et al., 2016; 2017b; Abid et al., 2017; Elhami et al., 2017; Hsu et al., 2017; Unnikrishnan et al., 2018; Tsakiris & Peng, 2019). The development of practical algorithms, however, falls far behind from the following three aspects:

• Most of the works are only applicable to linear regression models.

• Some of the existing algorithms are of very high computational complexity, and can only handle small number of data points in low dimensions (Elhami et al., 2017; Pananjady et al., 2017a; Tsakiris et al., 2018; Peng & Tsakiris, 2020). For example, Abid & Zou (2018) adopt an Expectation Maximization (EM) method where Metropolis-Hastings sampling is needed, which is not scalable. Other algorithms choose to optimize with respect to $w$ and $\pi$ in an alternating manner, e.g., alternating minimization in Abid et al. (2017). However, as there exists a strong interaction between $w$ and $\pi$, the optimization landscape of equation 1 is ill-conditioned. Therefore, these algorithms are not effective and often get stuck in local optima.

• Most of the works only consider the case where there exists an exact one-to-one correspondence between $\mathcal{D}_1$ and $\mathcal{D}_2$. For many more scenarios, however, these two datasets are not necessarily well aligned. For example, consider $\mathcal{D}_1$ and $\mathcal{D}_2$ collected from two separate databases, where the users overlap, but are not identical. As a result, there exists only partial one-to-one correspondence. A similar situation also happens to multiple-object tracking: Some objects may leave the scene in one frame, and new objects may enter the scene in subsequent frames. Therefore, not all objects in different frames can be perfectly matched. The RWOC problem with partial correspondence is known as robust-RWOC, or rRWOC (Varol & Nejatbakhsh, 2019), and is much less studied in existing literature.

To address these concerns, we propose a new computational framework – ROBOT (Regression withOut correspondence using Bilevel OptimizaTion). Specifically, we propose to formulate the regression without correspondence as a continuous optimization problem. Then by exploiting the interaction between the regression model and the data correspondence, we propose to develop a hypergradient approach based on differentiable programming techniques (Duchi et al., 2008; Luise et al., 2018). Our hypergradient approach views the data correspondence as an operator of the regression, i.e., for a given $w$, the optimal correspondence is

$$\widehat{\pi}(w) = \arg\min_{\pi} \mathcal{L}(w, \pi). \tag{2}$$

Accordingly, when applying gradient descent to (1), we need to find the gradient with respect to $w$ by differentiating through both the objective function $\mathcal{L}$ and the data correspondence $\widehat{\pi}(w)$. For simplicity, we refer as such a gradient to "hypergradient". Note that due to its discrete nature, $\widehat{\pi}(w)$ is actually not continuous in $w$. Therefore, such a hypergradient does not exist. To address this issue, we further propose to construct a smooth approximation of $\widehat{\pi}(w)$ by adding an additional regularizer to equation 2, and then we replace $\widehat{\pi}(w)$ with our proposed smooth replacement when computing the hyper gradient of $w$. Moreover, we also propose an efficient and scalable implementation of

hypergradient computation based on simple first order algorithms and implicit differentiation, which outperforms conventional automatic differentiation in terms of time and memory cost.

ROBOT can also be extended to the robust RWOC problem, where $\mathcal{D}_1$ and $\mathcal{D}_2$ are not necessarily exactly aligned, i.e., some data points in $\mathcal{D}_1$ may not correspond to any data point in $\mathcal{D}_2$. Specifically, we relax the constraints on the permutation $\pi(\cdot)$ (Liero et al., 2018) to automatically match related data points and ignore the unrelated ones.

At last, we conduct thorough numerical experiments to demonstrate the effectiveness of ROBOT. For RWOC (i.e., exact correspondence), we use several synthetic regression datasets and a real gated flow cytometry dataset, and we show that ROBOT outperforms baseline methods by significant margins. For robust RWOC (i.e., inexact correspondence), in addition to synthetic datasets, we consider a vision-based multiple-object tracking task, and then we show that ROBOT also achieves significant improvement over baseline methods.

**Notations**. Let $\|\cdot\|_2$ denote the $\ell_2$ norm of vectors, $\langle \cdot, \cdot \rangle$ the inner product of matrices, i.e., $\langle A, B \rangle = \sum_{i,j} A_{ij} B_{ij}$ for matrices $A$ and $B$. $a_{i:j}$ are the entries from index $i$ to index $j$ of vector $a$. Let $\mathbf{1}_n$ denote an $n$-dimensional vector of all ones. Denote $\frac{d(\cdot)}{d(\cdot)}$ the gradient of scalars, and $\nabla_{(\cdot)}(\cdot)$ the Jacobian of tensors. We denote $[v_1, v_2]$ the concatenation of two vectors $v_1$ and $v_2$. $\mathcal{N}(\mu, \sigma^2)$ is the Gaussian distribution with mean $\mu$ and variance $\sigma^2$.

## 2 ROBOT: A HYPERGRADIENT APPROACH FOR RWOC

We develop our hypergradient approach for RWOC. Specifically, we first introduce a continuous formulation equivalent to (1), and then propose a smooth bi-level relaxation with an efficient hypergradient descent algorithm.

### 2.1 EQUIVALENT CONTINUOUS FORMULATION

We propose a continuous optimization problem equivalent to (1). Specifically, we rewrite an equivalent form of (1) as follows,

$$\min_w \min_{S \in \mathbb{R}^{n \times n}} \mathcal{L}(w, S) = \langle C(w), S \rangle \quad \text{subject to } S \in \mathcal{P}, \tag{3}$$

where $\mathcal{P}$ denotes the set of all $n \times n$ permutation matrices, $C(w) \in \mathbb{R}^{n \times n}$ is the loss matrix with

$$C_{ij}(w) = \|y_i - f(x_i, z_j; w)\|_2^2.$$

Note that we can relax $\mathcal{S} \in \mathcal{P}$, which is the discrete feasible set of the inner minimization problem of (3), to a convex set, without affecting the optimality, as suggested by the next theorem.

**Proposition 1.** Given any $a \in \mathbb{R}^n$ and $b \in \mathbb{R}^m$, we define

$$\Pi(a, b) = \{A \in \mathbb{R}^{n \times m} : A\mathbf{1}_m = a, A^\top \mathbf{1}_n = b, A_{ij} \geq 0\}.$$

The optimal solution to the inner discrete minimization problem of (3) is also the optimal solution to the following continuous optimization problem,

$$\min_{S \in \mathbb{R}^{n \times n}} \langle C(w), S \rangle, \quad \text{s.t. } S \in \Pi(\mathbf{1}_n, \mathbf{1}_n). \tag{4}$$

This is a direct corollary of the Birkhoff-von Neumann theorem (Birkhoff, 1946; Von Neumann, 1953), and please refer to Appendix A for more details. Theorem 1 allows us to replace $\mathcal{P}$ in (3) with $\Pi(\mathbf{1}_n, \mathbf{1}_n)$, which is also known as the Birkhoff polytope[1](Ziegler, 2012). Accordingly, we obtain the following continuous formulation,

$$\min_w \min_{S \in \mathbb{R}^{n \times n}} \langle C(w), S \rangle \quad \text{subject to } S \in \Pi(\mathbf{1}_n, \mathbf{1}_n). \tag{5}$$

**Remark 1.** In general, equation 3 can be solved by linear programming algorithms (Dantzig, 1998).

### 2.2 CONVENTIONAL WISDOM: ALTERNATING MINIMIZATION

Conventional wisdom for solving (5) suggests to use alternating minimization (AM, Abid et al. (2017)). Specifically, at the $k$-th iteration, we first update $S$ by solving

$$S^{(k)} = \arg\min_{S \in \Pi(\mathbf{1}_n, \mathbf{1}_n)} \mathcal{L}(w^{(k-1)}, S),$$

and then given $S^{(k)}$, we update $w$ using gradient descent or exact minimization, i.e.,

$$w^{(k)} = w^{(k-1)} - \eta \nabla_w \mathcal{L}(w^{(k-1)}, S^{(k)}).$$

However, AM works poorly for solving (5) in practice. This is because $w$ and $S$ have a strong interaction throughout the iterations: A slight change to $w$ may lead to significant change to $S$. Therefore, the optimization landscape is ill-conditioned, and AM can easily get stuck at local optima.

---

[1]This is a common practice in integer programming (Marcus & Ree, 1959).

## 2.3 SMOOTH BI-LEVEL RELAXATION

To tackle the aforementioned computational challenge, we propose a hypergradient approach, which can better handle the interaction between $w$ and $S$. Specifically, we first relax (5) to a smooth bi-level optimization problem, and then we solve the relaxed bi-level optimization problem using the hypergradient descent algorithm.

We rewrite (5) as a smoothed bi-level optimization problem,

$$\min_w \mathcal{F}_\epsilon(w) = \langle C(w), S_\epsilon^*(w)\rangle, \text{ subject to } S_\epsilon^*(w) = \arg\min_{S\in\Pi(\mathbf{1}_n,\mathbf{1}_n)}\langle C(w), S\rangle + \epsilon H(S), \quad (6)$$

where $H(S) = \langle \log S, S\rangle$ is the entropy of $S$. The regularizer $H(S)$ in equation 6 alleviates the sensitivity of $S^*(w)$ to $w$. Note that if without such a regularizer, we solve

$$S^*(w) = \arg\min_{S\in\Pi(\mathbf{1}_n,\mathbf{1}_n)}\langle C(w), S\rangle. \quad (7)$$

The resulting $S^*(w)$ can be discontinuous in $w$. This is because $S^*(w)$ is the optimal solution of a linear optimization problem, and usually lies on a vertex of $\Pi(\mathbf{1}_n,\mathbf{1}_n)$. This means that if we change $w$, $S^*(w)$ either stays the same or jumps to another vertex of $\Pi(\mathbf{1}_n,\mathbf{1}_n)$. The jump makes $S^*(w)$ highly sensitive to $w$. To alleviate this issue, we propose to smooth $S^*(w)$ by adding an entropy regularizer to the lower level problem. The entropy regularizer enforces $S_\epsilon^*(w)$ to stay in the interior of $\Pi(\mathbf{1}_n,\mathbf{1}_n)$, and $S_\epsilon^*(w)$ changes smoothly with respect to $w$, as suggested by the following theorem.

**Theorem 1.** For any $\epsilon > 0$, $S_\epsilon^*(w)$ is differentiable, if the cost $C(w)$ is differentiable with respect to $w$. Consequently, the objective $\mathcal{F}_\epsilon(w) = \langle C(w), S_\epsilon^*(w)\rangle$ is also differentiable.

The proof is deferred to Appendix C. Note that (6) provides us a new perspective to interpret the relationship between $w$ and $S$. As can be seen from (6), $w$ and $S$ have different priorities: $w$ is the parameter of the leader problem, which is of the higher priority; $S$ is the parameter of the follower problem, which is of the lower priority, and can also be viewed as an operator of $w$ – denoted by $S_\epsilon^*(w)$. Accordingly, when we minimize (6) with respect to $w$ using gradient descent, we should also differentiate through $S_\epsilon^*$. We refer to such a gradient as "hypergradient" defined as follows,

$$\nabla_w \mathcal{F}_\epsilon(w) = \frac{\partial \mathcal{F}_\epsilon(w)}{\partial C(w)}\frac{\partial C(w)}{\partial w} + \frac{\partial \mathcal{F}_\epsilon(w)}{\partial S_\epsilon^*(w)}\frac{\partial S_\epsilon^*(w)}{\partial w} = \nabla_w \mathcal{L}(w, S) + \frac{\partial \mathcal{F}_\epsilon(w)}{\partial S_\epsilon^*(w)}\frac{\partial S_\epsilon^*(w)}{\partial w}.$$

We further examine the alternating minimization algorithm from the bi-level optimization perspective: Since $\nabla_w \mathcal{L}(w^{(k-1)}, S^{(k)})$ is not differentiable through $S^{(k)}$, AM is essentially using an inexact gradient. From a game-theoretic perspective[2], (6) defines a competition between the leader $w$ and the follower $S$. When using AM, $S$ only reacts to what $w$ has responded. In contrast, when using the hypergradient approach, the leader essentially recognizes the follower's strategy and reacts to what the follower is anticipated to response through $\frac{\partial \mathcal{F}_\epsilon(w)}{\partial S_\epsilon^*(w)}\frac{\partial S_\epsilon^*(w)}{\partial w}$. In this way, we can find a better descent direction for $w$.

**Remark 2.** We use a simple example of quadratic minimization to illustrative why we expect the bilevel optimization formulation in (6) to enjoy a benign optimization landscape. We consider a quadratic function

$$L(a_1, a_2) = a^\top P a + b^\top a, \quad (8)$$

where $a_1 \in \mathbb{R}^{d_1}$, $a_2 \in \mathbb{R}^{d_2}$, $a = [a_1, a_2]$, $P \in \mathbb{R}^{(d_1+d_2)\times(d_1+d_2)}$, $b \in \mathbb{R}^{d_1+d_2}$. Let $P = \rho\mathbf{1}_{d_1+d_2}\mathbf{1}_{d_1+d_2}^\top + (1-\rho)I_{d_1+d_2}$, where $I_{d_1+d_2}$ is the identity matrix, and $\rho$ is a constant. We solve the following bilevel optimization problem,

$$\min_{a_1} F(a_1) = L(a_1, a_2^*(a_1)) \quad \text{subject to } a_2^*(a_1) = \arg\min_{a_2} L(a_1, a_2) + \lambda\|a_2\|_2^2, \quad (9)$$

where $\lambda$ is a regularization coefficient. The next proposition shows that $\nabla^2 F(a_1)$ enjoys a smaller condition number than $\nabla_{a_1 a_1}^2 L(a_1, a_2)$, which corresponds to the problem that AM solves.

**Proposition 2.** Given $F$ defined in (9), we have

$$\frac{\lambda_{\max}(\nabla^2 F(a_1))}{\lambda_{\min}(\nabla^2 F(a_1))} = 1 + \frac{1-\rho+\lambda}{d_2\rho-\rho+\lambda+1}\cdot\frac{d_1\rho}{1-\rho} \quad \text{and} \quad \frac{\lambda_{\max}(\nabla_{a_1 a_1}^2 L(a_1, a_2))}{\lambda_{\min}(\nabla_{a_1 a_1}^2 L(a_1, a_2))} = 1 + \frac{d_1\rho}{1-\rho}.$$

The proof is deferred to Appendix B.1. As suggested by Proposition 2, $F(a_1)$ is much better-conditioned than $L(a_1, a_2)$ in terms of $a_1$ for high dimensional settings.

---

[2]The bilevel formulation can be viewed as a Stackelberg game.

## 2.4 Solving RWOC by Hypergradient Descent

We present how to solve (6) using our hypergradient approach. Specifically, we compute the "hypergradient" of $\mathcal{F}_\epsilon(w)$ based on the following theorem.

**Theorem 2.** The gradient of $\mathcal{F}_\epsilon$ with respect to $w$ is

$$\nabla_w \mathcal{F}_\epsilon(w) = \frac{1}{\epsilon} \sum_{i,j=1}^{n,n} \left( (1 - C_{ij}) S^*_{\epsilon,ij} + \sum_{h,\ell=1}^{n,n} C_{h\ell} S^*_{\epsilon,h\ell} P_{hij} + \sum_{h,\ell=1}^{n,n} C_{h\ell} S^*_{\epsilon,h\ell} Q_{\ell ij} \right) \nabla_w C_{ij}. \quad (10)$$

The definition of $P$ and $Q$ and the proof is deferred to Appendix C. Theorem 2 suggests that we first solve the lower level problem in (6),

$$S^*_\epsilon = \arg\min_{S \in \Pi(\mathbf{1}_n, \mathbf{1}_n)} \langle C(w), S \rangle + \epsilon H(S), \quad (11)$$

and then substitute $S^*_\epsilon$ into (10) to obtain $\nabla_w \mathcal{F}_\epsilon(w)$.

Note that the optimization problem in (11) can be efficiently solved by a variant of Sinkhorn algorithm (Cuturi, 2013; Benamou et al., 2015). Specifically, (11) can be formulated as an entropic optimal transport (EOT) problem (Monge, 1781; Kantorovich, 1960), which aims to find the optimal way to transport the mass from a categorical distribution with weight $\mu = [\mu_1, \ldots, \mu_n]^\top$ to another categorical distribution with weight $\nu = [\nu_1, \ldots, \nu_m]^\top$,

$$\Gamma^* = \arg\min_{\Gamma \in \Pi(\mu,\nu)} \langle M, \Gamma \rangle + \epsilon H(\Gamma),$$
$$\text{with } \Pi(\mu, \nu) = \{\Gamma \in \mathbb{R}^{n \times m} : \Gamma \mathbf{1}_m = \mu, \Gamma^\top \mathbf{1}_n = \nu, \Gamma_{ij} \geq 0\}, \quad (12)$$

where $M \in \mathbb{R}^{n \times m}$ is the cost matrix with $M_{ij}$ the transport cost. When we set the two categorical distributions as the empirical distribution of $\mathcal{D}_1$ and $\mathcal{D}_2$, respectively,

$$M = C(w) \quad \text{and} \quad \mu = \nu = \mathbf{1}_n/n,$$

one can verify that (12) is a scaled lower problem of (6), and their optimal solutions satisfies $S^*_\epsilon = n\Gamma^*$. Therefore, we can apply Sinkhorn algorithm to solve the EOT problem in equation 12: At the $\ell$-th iteration, we take

$$p^{(\ell+1)} = \frac{\mu}{Gq^{(\ell)}} \text{ and } q^{(\ell+1)} = \frac{\nu}{G^\top p^{(\ell+1)}}, \quad \text{where } q^{(0)} = \frac{1}{n}\mathbf{1}_n \text{ and } G_{ij} = \exp\left(\frac{-C_{ij}(w)}{\epsilon}\right),$$

$G \in \mathbb{R}^{n \times n}$, and the division here is entrywise. Let $p^*$ and $q^*$ denote the stationary points. Then we obtain $S^*_{\epsilon,ij} = np^*_i G_{ij} q^*_j$.

**Remark 3.** The Sinkhorn algorithm is iterative and cannot exactly solve (11) within finite steps. As the Sinkhorn algorithm is very efficient and attains linear convergence, it suffices to well approximate the gradient $\nabla_w \mathcal{F}_\epsilon(w)$ using the output inexact solution.

## 3 ROBOT for Robust Correspondence

We next propose a robust version of ROBOT to solve rRWOC (Varol & Nejatbakhsh, 2019). Note that in (6), the constraint $S \in \Pi(\mathbf{1}_n, \mathbf{1}_n)$ enforces a one-to-one matching between $\mathcal{D}_1$ and $\mathcal{D}_2$. For rRWOC, however, such an exact matching may not exist. For example, we have $n < m$, where $n = |\mathcal{D}_1|$, $m = |\mathcal{D}_2|$. Therefore, we need to relax the constraint on $S$.

Motivated by the connection between (6) and (12), we propose to solve the lower problem[3]:

$$(S^*_r(w), \bar{\mu}^*, \bar{\nu}^*) = \arg\min_{S \in \Pi(\bar{\mu},\bar{\nu})} \langle C(w), S \rangle + \epsilon H(S), \quad (13)$$
$$\text{subject to } \bar{\mu}^\top \mathbf{1}_n = n, \ \bar{\nu}^\top \mathbf{1}_m = m, \ \|\bar{\mu} - \mathbf{1}_n\|_2^2 \leq \rho_1, \ \|\bar{\nu} - \mathbf{1}_m\|_2^2 \leq \rho_2,$$

where $S^*_r(w) \in \mathbb{R}^{n \times m}$ denotes an inexact correspondence between $\mathcal{D}_1$ and $\mathcal{D}_2$. As can be seen in (13), we relax the marginal constraint $\Pi(\mathbf{1}, \mathbf{1})$ in (6) to $\Pi(\bar{\mu}, \bar{\nu})$, where $\bar{\mu}, \bar{\nu}$ are required to not deviate much from $\mathbf{1}$. Problem (13) relaxes the marginal constraints $\Pi(\mathbf{1}, \mathbf{1})$ in the original problem to $\Pi(\bar{\mu}, \bar{\nu})$, where $\bar{\mu}, \bar{\nu}$ are picked such that they do not deviate too much from $\mathbf{1}$[4]. Illustrative examples of the exact and robust alignments are provided in Figure 1.

---

[3]The idea is inspired by the marginal relaxation of optimal transport, first independently proposed by Kondratyev et al. (2016) and Chizat et al. (2018a), and later developed by Chizat et al. (2018c); Liero et al. (2018). Chizat et al. (2018b) share the same formulation as ours.

[4]Here we measure the deviation using the Euclidean distance, and more detailed discussions can be found in Appendix F

Computationally, (13) can be solved by taking the Sinkhorn iteration and the projected gradient iteration in an alternating manner (See more details in Appendix D). Given $S_r^*(w)$, we solve the upper level optimization in (6) to obtain $w^*$, i.e.,

$$w^* = \arg\min_w \langle C(w), S_r^*(w) \rangle.$$

Similar to the previous section, we use a first-order algorithm to solve this problem, and we derive explicit expressions for the update rules. See Appendix E for details.

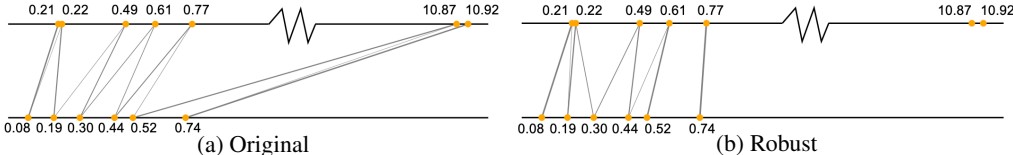

(a) Original        (b) Robust

Figure 1: *Illustrative example of exact (L) and robust (R) alignments. The robust alignment can drop potential outliers and only match data points close to each other.*

## 4 EXPERIMENT

We evaluate ROBOT and ROBOT-robust on both synthetic and real-world datasets, including flow cytometry and multi-object tracking. We first present numerical results and then we provide insights in the discussion section. Experiment details and auxiliary results can be found in Appendix G.

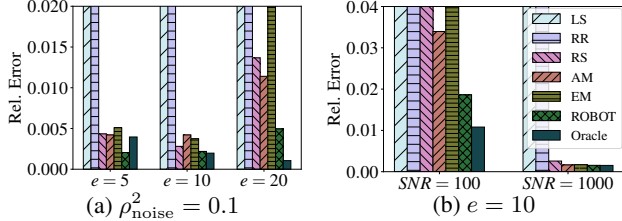

Figure 2: *Unlabeled sensing. Results are the mean over 10 runs.* SNR$= \|w\|_2^2/\rho_{\text{noise}}^2$ *is the signal-to-noise ratio.*

### 4.1 UNLABELED SENSING

**Data Generation**. We follow the unlabeled sensing setting (Tsakiris & Peng, 2019) and generate $n = 1000$ data points $\{(y_i, z_i)\}_{i=1}^n$, where $z_i \in \mathbb{R}^e$. Note here we take $d = 0$. We first generate $z_i, w \sim \mathcal{N}(\mathbf{0}_e, \mathbf{I}_e)$, and $\varepsilon_i \sim \mathcal{N}(0, \rho_{\text{noise}}^2)$. Then we compute $y_i = z_i^\top w + \varepsilon_i$. We randomly permute the order of 50% of $z_i$ so that we lose the $Z$-to-$Y$ correspondence. We generate the test set in the same way, only without permutation.

**Baselines and Training**. We consider the following scalable methods:

1. *Oracle*: Standard linear regression where no data are permuted.
2. *Least Squares (LS)*: Standard linear regression, i.e., treating the data as if they are not permuted.
3. *Alternating Minimization (AM*, Abid et al. (2017)): We iteratively solve the correspondence given $w$, and update $w$ using gradient descent with the correspondence.
4. *Stochastic EM* (Abid & Zou, 2018): A stochastic EM approach to recover the permutation.
5. *Robust Regression (RR*, Slawski & Ben-David (2019); Slawski et al. (2019a)). A two-stage block coordinate descent approach to discard outliers and fit regression models.
6. *Random Sample (RS*, Varol & Nejatbakhsh (2019)): A random sample consensus (RANSAC) approach to estimate $w$.

We initialize AM, EM and ROBOT using the output of RS with multi-start. We adopt a linear model $f(Z; w) = Z^\top w$. Models are evaluated by the relative error on the test set, i.e., error $= \sum_i (\widehat{y}_i - y_i)^2 / \sum_i (y_i - \bar{y})^2$, where $\widehat{y}_i$ is the predicted label, and $\bar{y}$ is the mean of $\{y_i\}$.

**Results**. We visualize the results in Figure 2. In all the experiments, ROBOT achieves better results than the baselines. Note that the relative error is larger for all methods except Oracle as the dimension and the noise increase. For low dimensional data, e.g., $e = 5$, our model achieves even better performance than Oracle. We have more discussions on using RS as initializations in Appendix G.5.

### 4.2 NONLINEAR REGRESSION

**Data Generation**. We mimic the scenario where the dataset is collected from different platforms. Specifically, we generate $n$ data points $\{(y_i, [x_i, z_i])\}_{i=1}^n$, where $x_i \in \mathbb{R}^d$ and $z_i \in \mathbb{R}^e$. We first generate $x_i \sim \mathcal{N}(\mathbf{0}_d, \mathbf{I}_d)$, $z_i \sim \mathcal{N}(\mathbf{0}_e, \mathbf{I}_e)$, $w \sim \mathcal{N}(\mathbf{0}_{d+e}, \mathbf{I}_{d+e})$, and $\varepsilon_i \sim \mathcal{N}(0, \rho_{\text{noise}}^2)$. Then we

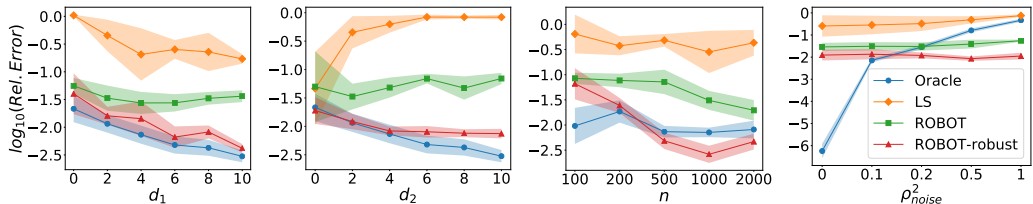

Figure 3: *Nonlinear regression. We use $n = 1000$, $d = 2$, $e = 3$, $\rho_{\mathrm{noise}}^2 = 0.1$ as defaults.*

compute $y_i = f([x_i, z_i]; w) + \varepsilon_i$. Next, we randomly permute the order of $\{z_i\}$ so that we lose the data correspondence. Here, $\mathcal{D}_1 = \{(x_i, y_i)\}$ and $\mathcal{D}_2 = \{z_j\}$ mimic two parts of data collected from two separate platforms. Since we are interested in the response on platform one, we treat all data from platform two, i.e., $\mathcal{D}_2$, as well as $80\%$ of data in $\mathcal{D}_1$ as the training data. The remaining data from $\mathcal{D}_1$ are the test data. Notice that we have different number of data on $\mathcal{D}_1$ and $\mathcal{D}_2$, i.e., the correspondence is not exactly one-to-one.

**Baselines and Training**. We consider a nonlinear function $f(X, Z; w) = \sum_{k=1}^{d} \sin\left([X, Z]_k w_k\right)$. In this case, we consider only two baselines — Oracle and LS, since the other baselines in the previous section are designed for linear models. We evaluate the regression models by the transport cost divided by $\sum_i (y_i - \bar{y})^2$ on the test set.

**Results**. As shown in Figure 3, ROBOT-robust consistently outperforms ROBOT and LS, demonstrating the effectiveness of our robust formulation. Moreover, ROBOT-robust achieves better performance than Oracle when the number of training data is large or when the noise level is high.

## 4.3 FLOW CYTOMETRY

In flow cytometry (FC), a sample containing particles is suspended in a fluid and injected into the flow cytometer, but the measuring instruments are unable to preserve the correspondence between the particles and the measurements. Different from FC, gated flow cytometry (GFC) uses "gates" to sort the particles into one of many bins, which provides partial ordering information since the measurements are provided individually for each bin. In practice, there are usually 3 or 4 bins.

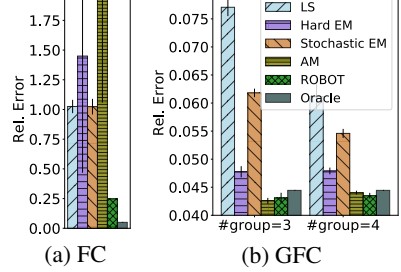

(a) FC     (b) GFC

Figure 4: *Relative error of different methods.*

**Settings**. We adopt the dataset from Knight et al. (2009). Following Abid et al. (2017), the outputs $y_i$'s are normalized, and we select the top 20 significant features by a linear regression on the top 1400 items in the dataset. We use $90\%$ of the data as the training data, and the remaining as test data. For ordinary FC, we randomly shuffle all the labels in the training set. For GFC, the training set is first sorted by the labels, and then divided into equal-sized groups, mimicking the sorting by gates process. The labels in each group are then randomly shuffled. To simulate gating error, $1\%$ of the data are shuffled across the groups. We compare ROBOT with Oracle, LS, Hard EM (a variant of Stochastic EM proposed in Abid & Zou (2018)), Stochastic EM, and AM. We use relative error on the test set as the evaluation metric.

**Results**. As shown in Figure 4, while AM achieves good performance on GFC when the number of groups is 3, it behaves poorly on the FC task. ROBOT, on the other hand, is efficient on both tasks.

## 4.4 MULTI-OBJECT TRACKING

In this section we extend our method to vision-based Multi-Object Tracking (MOT), a task with broad applications in mobile robotics and autonomous driving, to show the potential of applying RWOC to more real-world tasks. Given a video and the current frame, the goal of MOT is to predict the locations of the objects in the next frame. Specifically, object detectors (Felzenszwalb et al., 2009; Ren et al., 2015) first provide us the potential locations of the objects by their bounding boxes. Then, MOT aims to assign the bounding boxes to trajectories that describe the path of individual objects over time. Here, we formulate the current frame and the objects' locations in the current frame as $\mathcal{D}_2 = \{z_j\}$, while we treat the next frame and the locations in the next frame as $\mathcal{D}_1 = \{(x_i, y_i)\}$.

Table 1: *Experiment results on MOT.*

| Data | Method | MOTA↑ | MOTP↑ | IDF1↑ | MT↑ | ML↓ | FP↓ | FN↓ | IDS↓ |
|------|--------|-------|-------|-------|-----|-----|-----|-----|------|
| MOT17 (train) | ROBOT | **48.3** | 82.6 | 55.3 | 407 | 553 | 22,443 | 149,988 | 1,811 |
| | w/o ROBOT | 44.0 | 81.3 | 49.9 | 404 | 550 | 36,187 | 149,131 | 3,204 |
| MOT17 (dev) | ROBOT | **48.2** | 76.6 | 43.4 | 455 | 904 | 29,419 | 259,714 | 3,228 |
| | w/o ROBOT | 42.1 | 75.0 | 36.8 | 414 | 890 | 61,210 | 259,318 | 6,138 |
| | SORT | 43.1 | 77.8 | 39.8 | 295 | 997 | 28,398 | 287,582 | 4,852 |
| MOT20 (train) | ROBOT | **56.2** | 84.9 | 47.6 | 805 | 288 | 113,752 | 377,247 | 5,888 |
| | w/o ROBOT | 48.8 | 81.5 | 40.2 | 769 | 290 | 186,245 | 384,562 | 10,153 |
| MOT20 (dev) | ROBOT | **45.0** | 76.9 | 34.0 | 394 | 257 | 70,416 | 210,425 | 3,683 |
| | w/o ROBOT | 38.5 | 75.1 | 27.0 | 383 | 233 | 104,958 | 207,627 | 5,696 |
| | SORT | 42.7 | 78.5 | 45.1 | 208 | 326 | 27,521 | 264,694 | 4,470 |

Existing deep learning based MOT algorithms require large amounts of annotated data, i.e., the ground truth of the correspondence, during training. Different from them, our algorithm does not require the correspondence between $\mathcal{D}_1$ and $\mathcal{D}_2$, and all we need is the video. This task is referred to as *unsupervised MOT* (He et al., 2019).

**Related Works**. To the best of our knowledge, the only method that accomplishes unsupervised end-to-end learning of MOT is He et al. (2019). However, it targets tracking with low densities, e.g., Sprites-MOT, which is different from our focus.

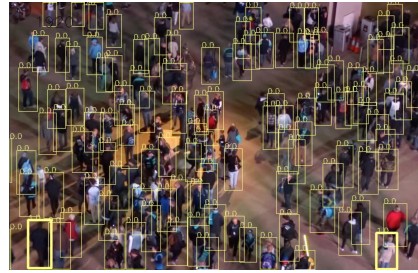

Figure 5: *One frame in MOT20 with detected bounding boxes in yellow.*

**Settings**. We adopt the MOT17 (Milan et al., 2016) and the MOT20 (Dendorfer et al., 2020) datasets. Scene densities of the two datasets are 31.8 and 170.9, respectively, which means the scenes are pretty crowded as we illustrated in Figure 5. We adopt the DPM detector (Felzenszwalb et al., 2009) on MOT17 and the Faster-RCNN detector (Ren et al., 2015) on MOT20 to provide us the bounding boxes. Inspired by Xu et al. (2019b), the cost matrix is computed as the average of the Euclidean center-point distance and the Jaccard distance between the bounding boxes,

$$C_{ij}(w) = \frac{1}{2}\left(\frac{\|c(f(z_j; w)) - c(y_i)\|_2}{\sqrt{H^2 + W^2}} + \mathcal{J}(f(z_j; w), y_i)\right),$$

where $c(\cdot)$ is the location of the box center, $H$ and $W$ are the height and the width of the video frame, and $\mathcal{J}(\cdot, \cdot)$ is the Jaccard distance defined as 1-IoU (Intersection-over-Union). We utilize the single-object tracking model SiamRPN[5] (Li et al., 2018) as our regression model $f$. We apply ROBOT-robust with $\rho_1 = \rho_2 = 10^{-3}$. See Appendix G for more detailed settings.

**Results**. We demonstrate the experiment results in Table 1, where the evaluation metrics follow Ristani et al. (2016). In the table, ↑ represents the higher the better, and ↓ represents the lower the better. ROBOT signifies the model trained by ROBOT-robust, and w/o ROBOT means the pretrained model in Li et al. (2018). The scores are improved significantly after training with ROBOT-robust.

We also include the scores of the SORT model (Bewley et al., 2016) obtained from the dataset platform. Different from SiamRPN and SiamRPN+ROBOT, SORT is a supervised learning model. As shown, our unsupervised training framework achieves comparable or even better performance.

## 5 DISCUSSION

**Sensitivity to initialization**. As stated in Pananjady et al. (2017b), obtaining the global optima of (1) is in general an NP-hard problem. Some "global" methods use global optimization techniques and have exponential complexity, e.g., Elhami et al. (2017), which is not applicable to large data. The other "local" methods only guarantee converge to local optima, and the convergence is very sensitive to initialization. Compared with existing "local" methods, our method is computationally efficient and greatly reduces the sensitivity to initialization.

---

[5]The initial weights of $f$ are obtained from https://github.com/foolwood/DaSiamRPN.

To demonstrate such an advantage, we run AM and ROBOT with 10 different initial solutions, and then we sort the results based on (a) the averaged residual on the training set, and (b) the relative prediction error on the test set. We plot the percentiles in Figure 6. Here we use fully shuffled data under the unlabeled sensing setting, and we set $n = 1000$, $e = 5$, $\rho_{\text{noise}}^2 = 0.1$, and $\epsilon = 10^{-2}$. We can see that ROBOT can find "good" solutions in 30% of the cases (The relative prediction error is smaller than 1), but AM is more sensitive to the initialization and cannot find "good" solutions.

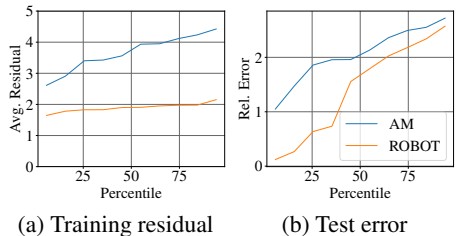

(a) Training residual    (b) Test error

Figure 6: *Results of different initialization of AM and ROBOT.*

**ROBOT v.s. Automatic Differentiation (AD).** Our algorithm computes the Jacobian matrix directly based on the KKT condition of the lower problem (11). An alternative approach to approximate the Jacobian is the automatic differentiation through the Sinkhorn iterations for updating $S$ when solving (11). As suggested by Figure 7 (a), running Sinkhorn iterations until conver-

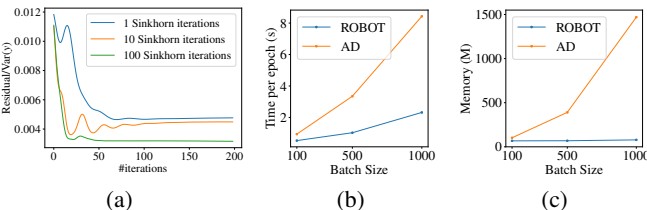

(a)    (b)    (c)

Figure 7: *The comparisons to AD. (a) Convergence under different number of Sinkhorn iterations of AD. (b) Time comparison. (c) Memory comparison.*

gence (200 Sinkhorn iterations) can lead to a better solution[6]. In order to apply AD, we need to store all the intermediate updates of all the Sinkhorn iterations. This require the memory usage to be proportional to the number of iterations, which is not necessarily affordable. In contrast, applying our explicit expression for the backward pass is memory-efficient. Moreover, we also observe that AD is much more time-consuming than our method. The timing performance and memory usage are shown in Figure 7 (b)(c), where we set $n = 1000$.

**Connection to EM.** Abid & Zou (2018) adopt an Expectation Maximization (EM) method for RWOC, where $S$ is modeled as a latent random variable. Then in the M-step, one maximizes the expected likelihood of the data over $S$. This method shares the same spirit as ours: We avoid updating $w$ using one single permutation matrix like AM. However, this method is very dependent on a good initialization. Specifically, if we randomly initialize $w$, the posterior distribution of $S$ in this iteration would be close to its prior, which is a uniform distribution. In this

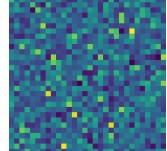

Figure 8: *Expected correspondence in EM.*

way, the follow-up update for $w$ is not informative. Therefore, the solution of EM would quickly converge to an undesired stationary point. Figure 8 illustrates an example of converged correspondence, where we adopt $n = 30, o = e = 1, d = 0$. For this reason, we initialize EM with good initial points, either by RS or AM throughout all experiments.

**Related works with additional constraints.** There is another line of research which improves the computational efficiency by solving variants of RWOC with additional constraints. Specifically, Haghighatshoar & Caire (2017); Rigollet & Weed (2018) assume an isotonic function (note that such an assumption may not hold in practice), and Shi et al. (2018); Slawski & Ben-David (2019); Slawski et al. (2019a;b); Varol & Nejatbakhsh (2019) assume only a small fraction of the correspondence is missing. Our method is also applicable to these problems, as long as the additional constraints can be adapted to the implicit differentiation.

**More applications of RWOC.** RWOC problems generally appear for two reasons. First, the measuring instruments are unable to preserve the correspondence. In addition to GFC and MOT, we list a few more examples: SLAM tracking (Thrun, 2007), archaeological measurements (Robinson, 1951), large sensor networks (Keller et al., 2009), pose and correspondence estimation (David et al., 2004), and the genome assembly problem from shotgun reads (Huang & Madan, 1999). Second, the data correspondence is masked for privacy reasons. For example, we want to build a recommender system for a new platform, borrowing user data from a mature platform.

---

[6]We remark that running one iteration sometimes cannot converge.

ACKNOWLEDGEMENT

This works is partially supported by NSF IIS-2008334. Hongteng Xu is supported in part by Beijing Outstanding Young Scientist Program (NO. BJJWZYJH012019100020098) and National Natural Science Foundation of China (No. 61832017). Xiaojing Ye is partially supported by NSF DMS-1925263. Hongyuan Zha is supported in part by a grant from Shenzhen Institute of Artificial Intelligence and Robotics for Society. We also appreciate the fruitful discussions with Bo Dai and Yan Li.

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

## A    CONNECTION BETWEEN OT AND RWOC

**Theorem 1.** Denote $\Pi(a, b) = \{S \in \mathbb{R}^{n \times m} : S\mathbf{1}_m = a, S^\top \mathbf{1}_n = b, S_{ij} \geq 0\}$ for any $a \in \mathbb{R}^n$ and $b \in \mathbb{R}^m$. Then at least one of the optimal solutions of the following problem lies in $\mathcal{P}$.

$$\min_{S \in \mathbb{R}^{n \times n}} \langle C(w), S \rangle, \quad \text{s.t. } S \in \Pi(\mathbf{1}_n, \mathbf{1}_n). \tag{14}$$

*Proof.* Denote the optimal solution of (14) as $Z^*$. As we mentioned earlier, this is a direct corollary of Birkhoff–von Neumann theorem (Birkhoff, 1946; Von Neumann, 1953). Specifically, Birkhoff–von Neumann theorem claims that the polytope $\Pi(\mathbf{1}_n, \mathbf{1}_n)$ is the convex hull of the set of $n \times n$ permutation matrices, and furthermore that the vertices of $\Pi(\mathbf{1}_n, \mathbf{1}_n)$ are precisely the permutation matrices.

On the other hand, (14) is a linear optimization problem. There would be at least one optimal solutions lies at the vertices given the problem is feasible. As a result, there would be at least one $Z^*$ being a permutation matrix. □

## B    TWO PERSPECTIVES OF THE MOTIVATIONS OF BILEVEL OPTIMIZATION

### B.1    FASTER CONVERGENCE

The bilevel optimization formulation has a better gradient descent iteration complexity than alternating minimization. To see this, consider a quadratic function $F(a_1, a_2) = a^\top P a + b^\top a$, where $a_1 \in \mathbb{R}^{d_1}$, $a_2 \in \mathbb{R}^{d_2}$, $a = [a_1^\top, a_2^\top]^\top \in \mathbb{R}^{(d_1+d_2)}$, $P \in \mathbb{R}^{(d_1+d_2) \times (d_1+d_2)}$, $b \in \mathbb{R}^{(d_1+d_2)}$. To further simplify the discussion, we assume $P = \rho \mathbf{1}_{(d_1+d_2)} \mathbf{1}_{(d_1+d_2)}^\top + (1 - \rho) I_{d_1+d_2}$, where $I_{d_1+d_2}$ is the identity matrix. Then we have the following proposition.

**Proposition 1.** Given $F$ defined in (9), we have

$$\frac{\lambda_{\max}(\nabla^2 F(a_1))}{\lambda_{\min}(\nabla^2 F(a_1))} = 1 + \frac{1 - \rho + \lambda}{1 - \rho} \frac{d_1 \rho}{d_2 \rho - \rho + \lambda + 1} \quad \text{and} \quad \frac{\lambda_{\max}(\nabla^2_{a_1 a_1} L(a_1, a_2))}{\lambda_{\min}(\nabla^2_{a_1 a_1} L(a_1, a_2))} = 1 + \frac{d_1 \rho}{1 - \rho}.$$

*Proof.* For alternating minimization, the Hessian for $a_1$ is a submatrix of $P$, i.e.,

$$H_{\text{AM}} = \rho \mathbf{1}_{d_1} \mathbf{1}_{d_1}^\top + (1 - \rho) I_{d_1},$$

whose condition number is

$$C_{\text{AM}} = 1 + \frac{d_1 \rho}{1 - \rho}.$$

We now compute the condition number for ROBOT. Denote

$$P = \begin{bmatrix} P_{11} & P_{12} \\ P_{21} & P_{22} \end{bmatrix}, \quad b = \begin{bmatrix} b_1 \\ b_2 \end{bmatrix},$$

where $P_{11} \in \mathbb{R}^{d_1 \times d_1}$, $P_{12} \in \mathbb{R}^{d_1 \times d_2}$, $P_{21} \in \mathbb{R}^{d_2 \times d_1}$, $P_{22} \in \mathbb{R}^{d_2 \times d_2}$, and $b_1 \in \mathbb{R}^{d_1}$, $b_2 \in \mathbb{R}^{d_2}$. ROBOT first minimize over $a_2$,

$$a_2^*(a_1) = \arg\min_{a_2} F(a_1, a_2) = -(P_{22} + \lambda I_{d_2})^{-1}(P_{21}a_1 + b_2/2).$$

Substituting $a_2^*(a_1)$ into $F(a_1, a_2)$, we can obtain the Hessian for $a_1$ is

$$H_{\text{ROBOT}} = P_{11} - P_{12}(P_{22} + \lambda I_{d_2})^{-1} P_{21}.$$

Using Sherman–Morrison formula, we can explicitly express $P_{22}^{-1}$ as

$$P_{22}^{-1} = \frac{1}{1 - \rho + \lambda} I_{d_2} - \frac{\rho}{(1 - \rho + \lambda)(1 - \rho + \lambda + \rho d_2)} \mathbf{1}_{d_2} \mathbf{1}_{d_2}^\top.$$

Substituting it into $H_{\text{ROBOT}}$,

$$H_{\text{ROBOT}} = P_{11} - P_{12} P_{22}^{-1} P_{21} = (1 - \rho) I_{d_1} + \left( \rho - \frac{d_2 \rho^2}{d_2 \rho - \rho + \lambda + 1} \right) \mathbf{1}_{d_1} \mathbf{1}_{d_1}^\top.$$

Therefore, the condition number is

$$C_{\text{ROBOT}} = 1 + \frac{1 - \rho + \lambda}{1 - \rho} \frac{d_1 \rho}{d_2 \rho - \rho + \lambda + 1}.$$

□

Note that $C_{\text{AM}}$ increases linearly with respect to $d_1$. Therefore, the optimization problem inevitably becomes ill-conditioned as dimension increase. In contrast, $C_{\text{ROBOT}}$ can stay in the same order of magnitude when $d_1$ and $d_2$ increase simultaneously.

Since the iteration complexity of gradient descent is proportional to the condition number (Bottou et al., 2018), ROBOT needs fewer iterations to converge than AM.

## C   DIFFERENTIABILITY

**Theorem 2.** For any $\epsilon > 0$, $S_\epsilon^*(w)$ is differentiable, as long as the cost $C(w)$ is differentiable with respect to $w$. As a result, the objective $\mathcal{L}_\epsilon(w) = \langle C(w), S_\epsilon^*(w) \rangle$ is also differentiable.

*Proof.* The proof is analogous to Xie et al. (2020).

We first prove the differentiability of $S_\epsilon^*(w)$. This part of proof mirrors the proof in Luise et al. (2018). By Sinkhorn's scaling theorem (Sinkhorn & Knopp, 1967),
$$S_\epsilon^*(w) = \text{diag}(e^{\frac{\xi^*(w)}{\epsilon}})e^{-\frac{C(w)}{\epsilon}}\text{diag}(e^{\frac{\zeta^*(w)}{\epsilon}}).$$
Therefore, since $C_{ij}(w)$ is differentiable, $\Gamma^{*,\epsilon}$ is differentiable if $(\xi^*(w), \zeta^*(w))$ is differentiable as a function of $w$.

Let us set
$$\mathcal{L}(\xi, \zeta; \mu, \nu, C) = \xi^T \mu + \zeta^T \nu - \epsilon \sum_{i,j=1}^{n,m} e^{-\frac{C_{ij}-\xi_i-\zeta_j}{\epsilon}}.$$
and recall that $(\xi^*, \zeta^*) = \arg\max_{\xi,\zeta} L(\xi, \zeta; \mu, \nu, C)$. The differentiability of $(\xi^*, \zeta^*)$ is proved using the Implicit Function theorem and follows from the differentiability and strict convexity in $(\xi^*, \zeta^*)$ of the function $\mathcal{L}$. $\qquad\square$

**Theorem 3.** Denoting $\mathcal{L}_\epsilon = \langle C(w), S_\epsilon^*(w) \rangle$. The gradient of $\mathcal{L}_\epsilon$ with respect to $w$ is
$$\nabla_w \mathcal{L}_\epsilon = \frac{1}{\epsilon} \sum_{i,j=1}^{n,n} \left( (1 - C_{ij})S_{\epsilon,ij}^* + \sum_{h,\ell=1}^{n,n} C_{h\ell}S_{\epsilon,h\ell}^* \frac{d\xi_h^*}{dC_{ij}} + \sum_{h,\ell=1}^{n,n} C_{h\ell}S_{\epsilon,h\ell}^* \frac{d\zeta_\ell^*}{dC_{ij}} \right) \nabla_w C_{ij}, \quad (15)$$
where $\begin{bmatrix} \nabla_C \xi^* \\ \nabla_C \zeta^* \end{bmatrix} = \begin{bmatrix} -H^{-1}D \\ \mathbf{0} \end{bmatrix}$ with $-H^{-1}D \in \mathbb{R}^{(2n-1)\times n \times n}, \mathbf{0} \in \mathbb{R}^{1\times n \times n}$,
$$D_{\ell ij} = \frac{1}{\epsilon} \begin{cases} \delta_{\ell i}S_{\epsilon,ij}^*, & \ell = 1, \cdots, n; \\ \delta_{\ell j}S_{\epsilon,ij}^*, & \ell = n+1, \cdots, 2n-1, \end{cases}$$
$$H^{-1} = -\epsilon \begin{bmatrix} (\text{diag}(\mu))^{-1} + (\text{diag}(\mu))^{-1}\bar{S}_\epsilon^* \mathcal{K}^{-1}\bar{S}_\epsilon^{*T}(\text{diag}(\mu))^{-1} & -(\text{diag}(\mu))^{-1}\bar{S}_\epsilon^* \mathcal{K}^{-1} \\ -\mathcal{K}^{-1}\bar{S}_\epsilon^{*T}(\text{diag}(\mu))^{-1} & \mathcal{K}^{-1} \end{bmatrix},$$
and $\mathcal{K} = \text{diag}(\bar{\nu}) - \bar{S}_\epsilon^{*T}(\text{diag}(\mu))^{-1}\bar{S}_\epsilon^*, \quad \bar{\nu} = \nu_{1:n-1}, \quad \bar{S}_\epsilon^* = S_{\epsilon,1:n,1:n-1}^*.$

*Proof.* This result is straightforward combining the Sinkhorn's scaling theorem and Theorem 3 in Xie et al. (2020). $\qquad\square$

## D   ALGORITHM OF THE FORWARD PASS FOR ROBOT-ROBUST

For better numerical stability, in practice we add two more regularization terms,
$$S_r^*(w), \bar{\mu}^*, \bar{\nu}^* = \arg\min_{S \in \Pi(\bar{\mu},\bar{\nu}), \, \bar{\mu}, \bar{\nu} \in \Delta_n} \langle C(w), S \rangle + \epsilon H(S) + \epsilon_1 h(\bar{\mu}) + \epsilon_2 h(\bar{\nu}), \quad (16)$$
$$\text{s.t.} \; \mathcal{F}(\bar{\mu}, \mu) \leq \rho_1, \; \mathcal{F}(\bar{\nu}, \nu) \leq \rho_2,$$
where $h(\bar{\mu}) = \sum_i \bar{\mu}_i \log \bar{\mu}_i$ is the entropy function for vectors. This can avoid the entries of $\bar{\mu}$ and $\bar{\nu}$ shrink to zeros when updated by gradient descent. We remark that since we have entropy term $H(S)$, the entries of $S$ would not be exactly zeros. Furthermore, we have $\bar{\mu} = S\mathbf{1}$ and $\bar{\mu} = S\mathbf{1}$. Therefore, theoretically the entries of $\bar{\mu}$ and $\bar{\nu}$ will not be zeros. We only add the two more entropy terms for numerical consideration. The detailed algorithm is in Algorithm 1. Although the algorithm is not guaranteed to converge to a feasible solution, in practice it usually converges to a good solution (Wang et al., 2015).

---

**Algorithm 1** Solving $S_r^*$ for robust matching

---

**Require:** $C \in \mathbb{R}^{m \times n}, \mu, \nu, K, \epsilon, L, \eta$

$\quad G_{ij} = e^{-\frac{C_{ij}}{\epsilon}}$

$\quad \bar{\mu} = \mu, \bar{\nu} = \nu$

$\quad b = \mathbf{1}_n$

$\quad$ **for** $l = 1, \cdots, L$ **do**

$\quad\quad a = \bar{\mu}/(Gb), b = \bar{\nu}/(G^T a)$

$\quad\quad \bar{\mu} = \bar{\mu} - \eta(e^{\frac{a}{\epsilon}} + \epsilon_1 * \log \bar{\mu}), \bar{\nu} = \bar{\nu} - \eta(e^{\frac{b}{\epsilon}} + \epsilon_2 * \log \bar{\nu})$

$\quad\quad \bar{\mu} = \max\{\bar{\mu}, 0\}, \bar{\nu} = \max\{\bar{\nu}, 0\}$

$\quad\quad \bar{\mu} = \bar{\mu}/(\bar{\mu}^\top \mathbf{1}), \bar{\nu} = \bar{\nu}/(\bar{\nu}^\top \mathbf{1})$

$\quad\quad$ **if** $\|\bar{\mu} - \mu\|_2^2 > \rho_1$ **then**

$\quad\quad\quad \bar{\mu} = \mu + \sqrt{\rho_1} \frac{\bar{\mu} - \mu}{\|\bar{\mu} - \mu\|_2}$

$\quad\quad$ **end if**

$\quad\quad$ **if** $\|\bar{\nu} - \nu\|_2^2 > \rho_2$ **then**

$\quad\quad\quad \bar{\nu} = \nu + \sqrt{\rho_2} \frac{\bar{\nu} - \nu}{\|\bar{\nu} - \nu\|_2}$

$\quad\quad$ **end if**

$\quad$ **end for**

$\quad S = \mathrm{diag}(a) \odot G \odot \mathrm{diag}(b)$

---

# E  ALGORITHM OF THE BACKWARD PASS FOR ROBOT-ROBUST

Since the derivation is tedious, we first summarize the outline of the derivation, then provide the detailed derivation.

## E.1  SUMMARY

Given $\bar{\mu}^*, \bar{\nu}^*, S_r^*(w)$, we compute the Jacobian matrix $dS_r^*(w)/dw$ using implicit differentiation and differentiable programming techinques. Specifically, the Lagrangian function of Problem (16) is

$$\mathcal{L} = \langle C, S \rangle + \epsilon H(S) + \epsilon_1 h(\bar{\mu}) + \epsilon_2 h(\bar{\nu}) - \xi^\top (\Gamma \mathbf{1}_m - \mu) - \zeta^\top (\Gamma^\top \mathbf{1}_n - \nu)$$
$$+ \lambda_1 (\bar{\mu}^\top \mathbf{1}_n - 1) + \lambda_2 (\bar{\nu}^\top \mathbf{1}_m - 1) + \lambda_3 (\|\bar{\mu} - \mu\|_2^2 - \rho_1) + \lambda_4 (\|\bar{\nu} - \nu\|_2^2 - \rho_2).$$

where $\xi$ and $\zeta$ are dual variables. The KKT conditions (Stationarity condition) imply that the optimal solution $\Gamma^{*,\epsilon}$ can be formulated using the optimal dual variables $\xi^*$ and $\zeta^*$ as,

$$S_r^* = \mathrm{diag}(e^{\frac{\xi^*}{\epsilon}}) e^{-\frac{C}{\epsilon}} \mathrm{diag}(e^{\frac{\zeta^*}{\epsilon}}). \tag{17}$$

By the chain rule, we have

$$\frac{dS_r^*}{dw} = \frac{dS_r^*}{dC}\frac{dC}{dw} = \left( \frac{\partial S_r^*}{\partial C} + \frac{\partial S_r^*}{\partial \xi^*}\frac{d\xi^*}{dC} + \frac{\partial S_r^*}{\partial \zeta^*}\frac{d\zeta^*}{dC} \right)\frac{dC}{dw}.$$

Therefore, we can compute $dS_r^*(w)/dw$ if we obtain $\frac{d\xi^*}{dC}$ and $\frac{d\zeta^*}{dC}$.

Substituting (17) into the Lagrangian function, at the optimal solutions we obtain

$$\mathcal{L} = \mathcal{L}(\xi^*, \zeta^*, \bar{\mu}^*, \bar{\nu}^*, \lambda_1^*, \lambda_2^*, \lambda_3^*, \lambda_4^*; C).$$

Denote $r^* = [(\xi^*)^\top, (\zeta^*)^\top, (\bar{\mu})^\top, (\bar{\nu})^\top, \lambda_1^*, \lambda_2^*, \lambda_3^*, \lambda_4^*]^\top$, and $\phi(r^*; C) = \partial \mathcal{L}(r^*; C)/\partial r^*$. At the optimal dual variable $r^*$, the KKT condition immediately yields $\phi(r^*; C) \equiv 0$. By the chain rule, we have

$$\frac{d\phi(r^*; C)}{dC} = \frac{\partial \phi(r^*; C)}{\partial C} + \frac{\partial \phi(r^*; C)}{\partial r^*}\frac{dr^*}{dC} = 0. \tag{18}$$

Rerranging terms, we obtain

$$\frac{dr^*}{dC} = -\left( \frac{\partial \phi(r^*; C)}{\partial r^*} \right)^{-1} \frac{\partial \phi(r^*; C)}{\partial C}. \tag{19}$$

Combining (17), (18), and (19), we can then obtain $dS_r^*(w)/dw$.

### E.2 DETAILS

Now we provide the detailed derivation for computing $dS_{\mathrm{r}}^*/dw$.

Since $S_{\mathrm{r}}^*$ is the optimal solution of an optimization problem, we can follow the implicit function theorem to solve for the closed-form expression of the gradient. Specifically, we adopt $\mathcal{F}(\bar{\mu}, \nu) = \sum_i (\bar{\mu}_i - \mu_i)^2$, and rewrite the optimization problem as

$$\min_{\bar{\mu}, \bar{\nu}, S} \langle C, S \rangle + \epsilon \sum_{ij} S_{ij}(\log S_{ij} - 1) + \epsilon_1 \sum_i \bar{\mu}_i(\log \bar{\mu}_i - 1) + \epsilon_2 \sum_j \bar{\nu}_j(\log \bar{\nu}_j - 1),$$

$$\text{s.t.,} \quad \sum_j S_{ij} = \bar{\mu}_i, \quad \sum_i S_{ij} = \bar{\nu}_j,$$

$$\sum_i \bar{\mu}_i = 1, \quad \sum_j \bar{\nu}_j = 1,$$

$$\sum_i (\bar{\mu}_i - \mu_i)^2 \leq \rho_1, \quad \sum_j (\bar{\nu}_j - \nu_j)^2 \leq \rho_2.$$

The Language of the above problem is

$$\mathcal{L}(C, S, \bar{\mu}, \bar{\nu}, \xi, \zeta, \lambda_1, \lambda_2, \lambda_3, \lambda_4)$$

$$= \langle C, S \rangle + \epsilon \sum_{ij} S_{ij}(\log S_{ij} - 1) + \epsilon_1 \sum_i \bar{\mu}_i(\log \bar{\mu}_i - 1) + \epsilon_2 \sum_j \bar{\nu}_j(\log \bar{\nu}_j - 1)$$

$$- \xi^\top (S\mathbf{1}_m - \bar{\mu}) - \zeta^\top (S^\top \mathbf{1}_n - \bar{\nu})$$

$$+ \lambda_1 (\sum_i \bar{\mu}_i - 1) + \lambda_2 (\sum_j \bar{\nu}_j - 1) + \lambda_3 (\sum_i (\bar{\mu}_i - \mu_i)^2 - \rho_1) + \lambda_4 (\sum_j (\bar{\nu}_j - \nu_j)^2 - \rho_2).$$

Easy to see that the Slater's condition holds. Denote

$$\mathcal{L}^* = \mathcal{L}(C, S_{\mathrm{r}}^*, \bar{\mu}^*, \bar{\nu}^*, \xi^*, \zeta^*, \lambda_1^*, \lambda_2^*, \lambda_3^*, \lambda_4^*).$$

Following the KKT conditions,

$$\frac{d\mathcal{L}^*}{dS_{\mathrm{r},ij}^*} = C_{ij} + \epsilon \log S_{\mathrm{r},ij}^* - \xi_i^* - \zeta_j^* = 0.$$

Therefore, $S_{\mathrm{r},ij}^* = e^{\frac{\xi_i^* + \zeta_j^* - C_{ij}}{\epsilon}}$. Then we have

$$\frac{dS_{\mathrm{r}}^*}{dw} = \left( \frac{\partial S_{\mathrm{r}}^*}{\partial C} + \frac{\partial S_{\mathrm{r}}^*}{\partial \xi^*} \frac{d\xi^*}{dC} + \frac{\partial S_{\mathrm{r}}^*}{\partial \zeta^*} \frac{d\zeta^*}{dC} \right) \frac{dC}{dw}.$$

So all we need to do is to compute $\frac{d\xi^*}{dC}$ and $\frac{d\zeta^*}{dC}$. Denote $F_{ij} = e^{\frac{\xi_i + \zeta_j - C_{ij}}{\epsilon}}$. Denote

$$\phi = \frac{d\mathcal{L}}{d\xi} = \bar{\mu} - F\mathbf{1}_m,$$

$$\psi = \frac{d\mathcal{L}}{d\zeta} = \bar{\nu} - F^\top \mathbf{1}_n,$$

$$p = \frac{d\mathcal{L}}{d\bar{\mu}} = \xi + \lambda_1 \mathbf{1}_n + 2\lambda_3(\bar{\mu} - \mu) + \epsilon_1 \log \bar{\mu},$$

$$q = \frac{d\mathcal{L}}{d\bar{\nu}} = \zeta + \lambda_2 \mathbf{1}_m + 2\lambda_4(\bar{\nu} - \nu) + \epsilon_2 \log \bar{\nu},$$

$$\chi_1 = \frac{d\mathcal{L}}{d\lambda_1} = \bar{\mu}^\top \mathbf{1}_n - 1,$$

$$\chi_2 = \frac{d\mathcal{L}}{d\lambda_2} = \bar{\nu}^\top \mathbf{1}_m - 1,$$

$$\chi_3 = \lambda_3(\|\bar{\mu} - \mu\|_2^2 - \rho_1),$$

$$\chi_4 = \lambda_4(\|\bar{\nu} - \nu\|_2^2 - \rho_2).$$

Denote $\chi = [\chi_1, \chi_2, \chi_3, \chi_4]$, and $\lambda = [\lambda_1, \lambda_2, \lambda_3, \lambda_4]$. Following the KKT conditions, we have

$$\phi = 0, \psi = 0, p = 0, q = 0, \chi = 0,$$

at the optimal solutions. Therefore, for the optimal solutions we have

$$\frac{d\phi}{dC} = \frac{\partial\phi}{\partial C} + \frac{\partial\phi}{\partial\xi^*}\frac{d\xi^*}{dC} + \frac{\partial\phi}{\partial\zeta^*}\frac{d\zeta^*}{dC} + \frac{\partial\phi}{\partial\bar{\mu}^*}\frac{d\bar{\mu}^*}{dC} + \frac{\partial\phi}{\partial\bar{\nu}^*}\frac{d\bar{\nu}^*}{dC} + \frac{\partial\phi}{\partial\lambda^*}\frac{d\lambda^*}{dC} = 0,$$

$$\frac{d\psi}{dC} = \frac{\partial\psi}{\partial C} + \frac{\partial\psi}{\partial\xi^*}\frac{d\xi^*}{dC} + \frac{\partial\psi}{\partial\zeta^*}\frac{d\zeta^*}{dC} + \frac{\partial\psi}{\partial\bar{\mu}^*}\frac{d\bar{\mu}^*}{dC} + \frac{\partial\psi}{\partial\bar{\nu}^*}\frac{d\bar{\nu}^*}{dC} + \frac{\partial\psi}{\partial\lambda^*}\frac{d\lambda^*}{dC} = 0,$$

$$\frac{dp}{dC} = \frac{\partial p}{\partial C} + \frac{\partial p}{\partial\xi^*}\frac{d\xi^*}{dC} + \frac{\partial p}{\partial\zeta^*}\frac{d\zeta^*}{dC} + \frac{\partial p}{\partial\bar{\mu}^*}\frac{d\bar{\mu}^*}{dC} + \frac{\partial p}{\partial\bar{\nu}^*}\frac{d\bar{\nu}^*}{dC} + \frac{\partial p}{\partial\lambda^*}\frac{d\lambda^*}{dC} = 0,$$

$$\frac{dq}{dC} = \frac{\partial q}{\partial C} + \frac{\partial q}{\partial\xi^*}\frac{d\xi^*}{dC} + \frac{\partial q}{\partial\zeta^*}\frac{d\zeta^*}{dC} + \frac{\partial q}{\partial\bar{\mu}^*}\frac{d\bar{\mu}^*}{dC} + \frac{\partial q}{\partial\bar{\nu}^*}\frac{d\bar{\nu}^*}{dC} + \frac{\partial q}{\partial\lambda^*}\frac{d\lambda^*}{dC} = 0$$

$$\frac{d\chi}{dC} = \frac{\partial\chi}{\partial C} + \frac{\partial\chi}{\partial\xi^*}\frac{d\xi^*}{dC} + \frac{\partial\chi}{\partial\zeta^*}\frac{d\zeta^*}{dC} + \frac{\partial\chi}{\partial\bar{\mu}^*}\frac{d\bar{\mu}^*}{dC} + \frac{\partial\chi}{\partial\bar{\nu}^*}\frac{d\bar{\nu}^*}{dC} + \frac{\partial\chi}{\partial\lambda^*}\frac{d\lambda^*}{dC} = 0.$$

Therefore, we have

$$
\begin{bmatrix} \frac{d\xi^*}{dC} \\ \frac{d\zeta^*}{dC} \\ \frac{d\bar{\mu}^*}{dC} \\ \frac{d\bar{\nu}^*}{dC} \\ \frac{d\lambda^*}{dC} \end{bmatrix} = -
\begin{bmatrix} \frac{\partial\phi}{\partial\xi^*} & \frac{\partial\phi}{\partial\zeta^*} & \frac{\partial\phi}{\partial\bar{\mu}^*} & \frac{\partial\phi}{\partial\bar{\nu}^*} & \frac{\partial\phi}{\partial\lambda^*} \\ \frac{\partial\psi}{\partial\xi^*} & \frac{\partial\psi}{\partial\zeta^*} & \frac{\partial\psi}{\partial\bar{\mu}^*} & \frac{\partial\psi}{\partial\bar{\nu}^*} & \frac{\partial\psi}{\partial\lambda^*} \\ \frac{\partial p}{\partial\xi^*} & \frac{\partial p}{\partial\zeta^*} & \frac{\partial p}{\partial\bar{\mu}^*} & \frac{\partial p}{\partial\bar{\nu}^*} & \frac{\partial p}{\partial\lambda^*} \\ \frac{\partial q}{\partial\xi^*} & \frac{\partial q}{\partial\zeta^*} & \frac{\partial q}{\partial\bar{\mu}^*} & \frac{\partial q}{\partial\bar{\nu}^*} & \frac{\partial q}{\partial\lambda^*} \\ \frac{\partial\chi}{\partial\xi^*} & \frac{\partial\chi}{\partial\zeta^*} & \frac{\partial\chi}{\partial\bar{\mu}^*} & \frac{\partial\chi}{\partial\bar{\nu}^*} & \frac{\partial\chi}{\partial\lambda^*} \end{bmatrix}^{-1}
\begin{bmatrix} \frac{\partial\phi}{\partial C} \\ \frac{\partial\psi}{\partial C} \\ \frac{\partial p}{\partial C} \\ \frac{\partial q}{\partial C} \\ \frac{\partial\chi}{\partial C} \end{bmatrix}.
$$

After some derivation, we have

$$
\begin{bmatrix} \frac{d\xi^*}{dC} \\ \frac{d\zeta^*}{dC} \\ \frac{d\bar{\mu}^*}{dC} \\ \frac{d\bar{\nu}^*}{dC} \\ \frac{d\lambda_1^*}{dC} \\ \frac{d\lambda_2^*}{dC} \\ \frac{d\lambda_3^*}{dC} \\ \frac{d\lambda_4^*}{dC} \end{bmatrix} = -
\begin{bmatrix}
-\frac{1}{\epsilon}\mathrm{diag}(\bar{\mu}) & -\frac{1}{\epsilon}S_r^* & \boldsymbol{I}_n & \boldsymbol{0} & \boldsymbol{0} & \boldsymbol{0} & \boldsymbol{0} & \boldsymbol{0} \\
-\frac{1}{\epsilon}(S_r^*)^\top & -\frac{1}{\epsilon}\mathrm{diag}(\bar{\nu}) & \boldsymbol{0} & \boldsymbol{I}_m & \boldsymbol{0} & \boldsymbol{0} & \boldsymbol{0} & \boldsymbol{0} \\
\boldsymbol{I}_n & \boldsymbol{0} & 2\lambda_3\boldsymbol{I}_n+\mathrm{diag}(\frac{\epsilon_1}{\bar{\mu}}) & \boldsymbol{0} & \boldsymbol{1}_n & \boldsymbol{0} & 2(\bar{\mu}-\mu) & \boldsymbol{0} \\
\boldsymbol{0} & \boldsymbol{I}_m & \boldsymbol{0} & 2\lambda_4\boldsymbol{I}_m+\mathrm{diag}(\frac{\epsilon_2}{\bar{\nu}}) & \boldsymbol{0} & \boldsymbol{1}_m & \boldsymbol{0} & 2(\bar{\nu}-\nu) \\
\boldsymbol{0} & \boldsymbol{0} & \boldsymbol{1}_n^\top & \boldsymbol{0} & 0 & 0 & 0 & 0 \\
\boldsymbol{0} & \boldsymbol{0} & \boldsymbol{0} & \boldsymbol{1}_m^\top & 0 & 0 & 0 & 0 \\
\boldsymbol{0} & \boldsymbol{0} & 2\lambda_3(\bar{\mu}-\mu)^\top & \boldsymbol{0} & 0 & 0 & \|\bar{\mu}-\mu\|_2^2-\rho_1 & 0 \\
\boldsymbol{0} & \boldsymbol{0} & \boldsymbol{0} & 2\lambda_4(\bar{\nu}-\nu)^\top & 0 & 0 & 0 & \|\bar{\nu}-\nu\|_2^2-\rho_2
\end{bmatrix}^{-1}
\begin{bmatrix} \frac{\partial\phi}{\partial C} \\ \frac{\partial\psi}{\partial C} \\ \boldsymbol{0} \\ \boldsymbol{0} \\ 0 \\ 0 \\ 0 \\ 0 \end{bmatrix},
$$

and

$$\frac{\partial\phi_h}{\partial C_{ij}} = \frac{1}{\epsilon}\delta_{hi}S_{ij}, \forall h=1,\cdots,n, \quad i=1,\cdots,n, \quad j=1,\cdots,m$$

$$\frac{\partial\psi_\ell}{\partial C_{ij}} = \frac{1}{\epsilon}\delta_{\ell j}S_{ij}, \forall \ell=1,\cdots,m-1, \quad i=1,\cdots,n, \quad j=1,\cdots,m.$$

To efficiently solve for the inverse in the above equations, we denote

$$
A = \begin{bmatrix}
-\frac{1}{\epsilon}\mathrm{diag}(\bar{\mu}) & -\frac{1}{\epsilon}S_r^* & \boldsymbol{I}_n & \boldsymbol{0} \\
-\frac{1}{\epsilon}(S_r^*)^\top & -\frac{1}{\epsilon}\mathrm{diag}(\bar{\nu}) & \boldsymbol{0} & \boldsymbol{I}_m \\
\boldsymbol{I}_n & \boldsymbol{0} & 2\lambda_3\boldsymbol{I}_n+\mathrm{diag}(\frac{\epsilon_1}{\bar{\mu}}) & \boldsymbol{0} \\
\boldsymbol{0} & \boldsymbol{I}_m & \boldsymbol{0} & 2\lambda_4\boldsymbol{I}_m+\mathrm{diag}(\frac{\epsilon_2}{\bar{\nu}})
\end{bmatrix},
$$

$$
B_1 = \begin{bmatrix} \boldsymbol{1}_n & \boldsymbol{0} & 2(\bar{\mu}-\mu) & \boldsymbol{0} \\ \boldsymbol{0} & \boldsymbol{1}_m & \boldsymbol{0} & 2(\bar{\nu}-\nu) \end{bmatrix},
$$

$$
C_1 = \begin{bmatrix} \boldsymbol{1}_n^\top & \boldsymbol{0} \\ \boldsymbol{0} & \boldsymbol{1}_m^\top \\ 2\lambda_3(\bar{\mu}-\mu)^\top & \boldsymbol{0} \\ \boldsymbol{0} & 2\lambda_4(\bar{\nu}-\nu)^\top \end{bmatrix},
$$

$$D = \begin{bmatrix} 0 & 0 & 0 & 0 \\ 0 & 0 & 0 & 0 \\ 0 & 0 & \|\bar{\mu} - \mu\|_2^2 - \rho_1 & 0 \\ 0 & 0 & 0 & \|\bar{\nu} - \nu\|_2^2 - \rho_2 \end{bmatrix}.$$

We first $A^{-1}$ using the rules for inverting a block matrix,

$$A^{-1} = \begin{bmatrix} K & -KL \\ -LK & L + LKL \end{bmatrix} =: \begin{bmatrix} A_1 & A_2 \\ A_3 & A_4 \end{bmatrix}$$

where

$$L = \begin{bmatrix} 2\lambda_3 \boldsymbol{I}_n + \mathrm{diag}(\frac{\epsilon_1}{\bar{\mu}}) & \mathbf{0} \\ \mathbf{0} & 2\lambda_4 \boldsymbol{I}_m + \mathrm{diag}(\frac{\epsilon_1}{\bar{\nu}}) \end{bmatrix}^{-1}, \quad K = \left( \frac{1}{\epsilon} \begin{bmatrix} \mathrm{diag}(\bar{\mu}) & S_{\mathrm{r}}^* \\ (S_{\mathrm{r}}^*)^\top & \mathrm{diag}(\bar{\nu}) \end{bmatrix} + L \right)^{-1}.$$

Then using the rules of inverting a block matrix again, we have

$$\begin{bmatrix} \frac{d\xi^*}{dC} \\ \frac{d\zeta_*}{dC} \end{bmatrix} = (A_1 + A_2 B_1 (D - C_1 A_4 B_1)^{-1} C_1 A_3) \begin{bmatrix} \frac{\partial\phi}{\partial C} \\ \frac{\partial\psi}{\partial C} \end{bmatrix}.$$

Therefore, the bottleneck of computation is the inverting step in computing $K$. Note $L$ is a diagonal matrix, we can further lower the computation cost by applying the rules for inverting a block matrix again. The value of $\lambda_3$ and $\lambda_4$ can be estimated from the fact $p = 0, q = 0$. We detail the algorithm in Algorithm 2.

---

**Algorithm 2** Computing the gradient for $w$

---

**Require:** $C \in \mathbb{R}^{m \times n}, \mu, \nu, \epsilon, \frac{dC}{dw}$

Run forward pass to get $S = S_{\mathrm{r}}^*, \bar{\mu}, \bar{\nu}, \xi, \zeta$

$x_1 = \sum_{i=1}^{\lceil n/2 \rceil} (\bar{\mu}_i - \mu_i), x_2 = \sum_{i=\lceil n/2 \rceil}^n (\bar{\mu}_i - \mu_i), b_1 = -\sum_{i=1}^{\lceil n/2 \rceil} \xi_i, b_2 = -\sum_{i=\lceil n/2 \rceil}^n \xi_i$

$[\lambda_1, \lambda_3]^\top = [\lceil n/2 \rceil, x_1; n - \lceil n/2 \rceil, x_2]^{-1} [b1, b2]^\top$

$x_1 = \sum_{j=1}^{\lceil m/2 \rceil} (\bar{\nu}_j - \nu_j), x_2 = \sum_{j=\lceil m/2 \rceil}^m (\bar{\nu}_j - \nu_j), b_1 = -\sum_{j=1}^{\lceil m/2 \rceil} \zeta_j, b_2 = -\sum_{j=\lceil m/2 \rceil}^m \zeta_j$

$[\lambda_2, \lambda_4]^\top = [\lceil m/2 \rceil, x_1; m - \lceil m/2 \rceil, x_2]^{-1} [b1, b2]^\top$

$\bar{\mu} = \bar{\mu} + \epsilon (2\lambda_3 \mathbf{1}_n + \frac{\epsilon_1}{\bar{\mu}})^{-1}, \bar{\nu} = \bar{\nu} + \epsilon (2\lambda_4 \mathbf{1}_m + \frac{\epsilon_2}{\bar{\nu}})^{-1}$

$\bar{\nu}' = \bar{\nu}[:-1], S' = S[:,:-1]$

$\mathcal{K} \leftarrow \mathrm{diag}(\bar{\nu}') - (S')^\top (\mathrm{diag}(\bar{\mu}))^{-1} S'$

$H_1 \leftarrow (\mathrm{diag}(\bar{\mu}))^{-1} + (\mathrm{diag}(\bar{\mu}))^{-1} S' \mathcal{K}^{-1} (S')^\top (\mathrm{diag}(\bar{\mu}))^{-1}$

$H_2 \leftarrow -(\mathrm{diag}(\bar{\mu}))^{-1} S' \mathcal{K}^{-1}$

$H_3 \leftarrow (H_2)^\top$

$H_4 \leftarrow \mathcal{K}^{-1}$

Pad $H_2$ to be $[n, m]$ with value 0

Pad $H_3$ to be $[m, n]$ with value 0

Pad $H_4$ to be $[m, m]$ with value 0

$L = \mathrm{diag}([\epsilon (2\lambda_3 \mathbf{1}_n + \frac{\epsilon_1}{\bar{\mu}})^{-1}, \epsilon (2\lambda_4 \mathbf{1}_m + \frac{\epsilon_2}{\bar{\nu}})^{-1}])$

$A1 = [H_1, H_2; H_3, H_4]$

$A_2 = -A_1 \cdot L$

$A_3 = A_2^\top$

$A_4 = L + L \cdot A_1 \cdot L$

$E = A_1 + A_2 \cdot B1 (D - C \cdot A_4 \cdot B)^{-1} C \cdot A_3$, where $B1, C_1, D$ defined above

$[J_1, J_2; J_3, J_4] = E$, where $J_1 \in \mathbb{R}^{n \times n}, J_2 \in \mathbb{R}^{n \times m}, J_3 \in \mathbb{R}^{m \times n}, J_4 \in \mathbb{R}^{m \times m}$

$[\frac{d\xi^*}{dC}]_{nij} \leftarrow [J_1]_{ni} S_{ij} + [J_2]_{nj} S_{ij}$

$[\frac{d\zeta^*}{dC}]_{mij} \leftarrow [J_3]_{mi} S_{ij} + [J_4]_{mj} S_{ij}$

Pad $\frac{d\zeta^*}{dC}$ to be $[m, n, m]$ with value 0

$[\frac{d\mathcal{L}}{dC}]_{ij} \leftarrow \frac{1}{\epsilon} (-C_{ij} S_{ij} + \sum_{n,m} C_{nm} S_{nm} [\frac{da^*}{dC}]_{nij} + \sum_{n,m} C_{nm} S_{nm} [\frac{db^*}{dC}]_{mij}) + S_{ij}$

**return** $\dfrac{d\mathcal{L}}{dC} \dfrac{dC}{dw}$

---

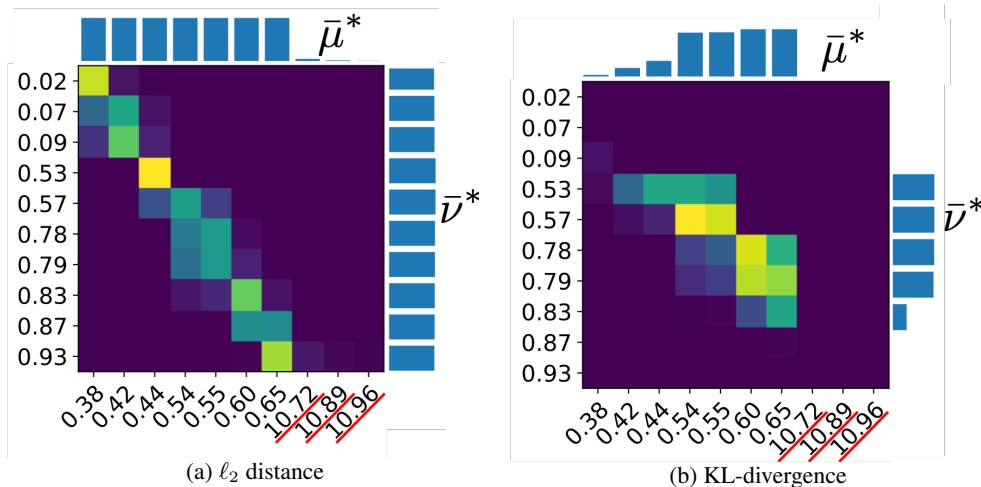

(a) $\ell_2$ distance  (b) KL-divergence

Figure 9: *Illustration with different choice of $\mathcal{F}$.*

## F  DIFFERENT FORMS OF MARGINAL RELAXATION

In this paper we adopt $\mathcal{F}$ to be the Euclidean distance. This is because this choice provides an OT plan that fits our intuition – the data points with significantly larger transportation cost should not be considered. Figure 9 shows an illustration. Here, the input distributions are the empirical distributions of the scalars on the left and the bottom. Notice that there are three support points in $\mu$ that are far away from others, i.e., $10.72, 10.89, 10.96$. In Figure 9 (a), the optimal solution $\Gamma_r^*$ automatically ignores them, matching only the rest of the scalars. One alternative choice of $\mathcal{F}$ is the Kullback–Leibler (KL) divergence (Chizat et al., 2018b), whose resulted formulation possesses an efficient algorithm for the forward pass, and the differentiability for the backward pass. We do not adopt it because the OT plan generated by this choice does not fit out intuition: As shown in Figure 9 (b), the OT plan tends to ignore the points that are away from the mean, even with a very small $\rho_1$ and $\rho_2$. For both figures, we adopt $\epsilon = 10^{-5}$.

## G  MORE ON EXPERIMENTS

### G.1  UNLABELED SENSING

We now provide more training details for experiments in Section 4.1. Here, AM and ROBOT is trained with batch size 500 and learning rate $10^{-4}$ for $2,000$ iterations. For the Sinkhorn algorithm in ROBOT we set $\epsilon = 10^{-4}$. We run RS for $2 \times 10^5$ iterations with inlier threshold as $10^{-2}$. Other settings for the hyper-parameters in the baselines follows the default settings of their corresponding papers.

### G.2  NONLINEAR REGRESSION

For the nonlinear regression experiment in Section 4.2, ROBOT and ROBOT-robust is trained with learning rate $10^{-4}$ for 80 iterations. For $n = 100, 200, 500, 1000, 2000$, we set batch size $10, 30, 50, 100, 300$, respectively. We set $\epsilon = 10^{-4}$ for the Sinkhorn algorithm in ROBOT. For Oracle and LS, we perform ordinary regression model and ensure convergence, i.e., learning rate $5 \times 10^{-2}$ for 100 iterations.

### G.3  FLOW CYTOMETRY

We provide more details for the Flow Cytometry experiment in Section 4.3. In the FC seting, ROBOT is trained with batch size 1260 and learning rate $10^{-4}$ for 80 iterations. In the GFC seting, ROBOT is trained with batch size 1260 and learning rate $6 \times 10^{-4}$ for 60 iterations. We set $\epsilon = 10^{-4}$ for the Sinkhorn algorithm in ROBOT. Other settings for the hyper-parameters in the baselines follows the default settings of their corresponding papers. EM is initialized by AM.

### G.4 MULTI-OBJECT TRACKING

For the MOT experiments in Section 4.4, the reported results of MOT17 (train) and MOT17 (dev) is trained on MOT17 (train), and the reported results of MOT20 (train) and MOT20 (dev) is trained on MOT20 (train). Each model is trained for 1 epoch. We adopt Adam optimizer with learning rate$= 10^{-5}$, $\epsilon = 10^{-4}$, and $\eta = 10^{-3}$. To track the birth and death of the tracks, we adapt the inference code of Xu et al. (2019b).

### G.5 COMBINATION WITH RS

As suggested in Figure 2, although RS cannot perform well itself, retraining the output of RS using our algorithms increases the performance by a large margin. To show that combining RS and ROBOT can achieve better results than RS alone, we compare the following two cases: i). Subsample $2 \times 10^5$ times using RS; ii). Subsample $10^5$ times us-

Table 2: *Pairwise comparisons between RS alone and the combination of RS and ROBOT. The relative error ratio is the ratio of the relative errors of RS alone and RS+ROBOT combination. Ratios larger than 1 suggest that RS performs worse than RS+ROBOT combination.*

| Proportion | 25% | 50% | 75% |
|---|---|---|---|
| Rel. error ratio | $1.04 \pm 0.20$ | $1.29 \pm 0.32$ | $1.27 \pm 0.34$ |

ing RS followed by ROBOT for 50 training steps. The result is shown in Table 2. For a larger permutation proportion, RS alone cannot perform as well as RS+ROBOT combination. Here, we have 10 runs for each proportion. We adopt SNR$= 100$, $d = 5$ for data, and $\epsilon = 10^{-4}$, learning rate $10^{-4}$ for ROBOT training.

### G.6 THE EFFECT OF $\rho_1$ AND $\rho_2$

We visualize $S_\mathrm{r}^*$ computed from the robust optimal transport problem in Figure 10. The two input distributions are Unif$(0, 2)$ and Unif$(0, 1)$. We can see that with large enough $\rho_1$ and $\rho_2$, Unif$(0, 1)$ would be aligned with the first half of Unif$(0, 2)$.

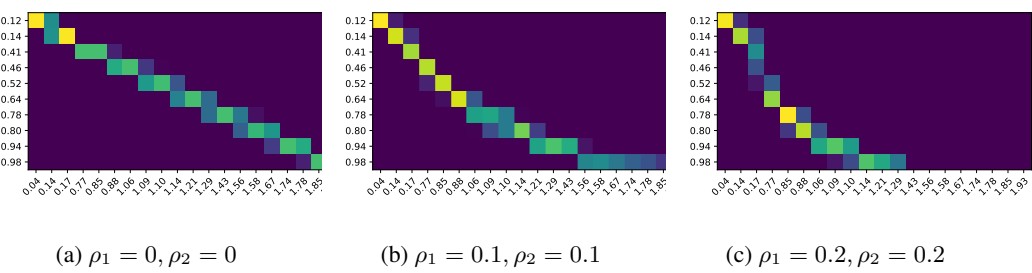

(a) $\rho_1 = 0, \rho_2 = 0$      (b) $\rho_1 = 0.1, \rho_2 = 0.1$      (c) $\rho_1 = 0.2, \rho_2 = 0.2$

Figure 10: Computed $S^*$ for robust optimal transport problem.

### G.7 COMPARISON OF RESIDUALS IN LINEAR REGRESSION

**Settings**. We generate $n$ data points $\{(y_i, [x_i, z_i])\}_{i=1}^n$, where $x_i \in \mathbb{R}^d$ and $z_i \in \mathbb{R}^e$. We first generate $x_i \sim \mathcal{N}(\mathbf{0}_d, \mathbf{I}_d)$, $z_i \sim \mathcal{N}(\mathbf{0}_e, \mathbf{I}_e)$, $w \sim \mathcal{N}(\mathbf{0}_{d+e}, \mathbf{I}_{d+e})$, and $\varepsilon_i \sim \mathcal{N}(0, \rho_{\mathrm{noise}}^2)$. Then we compute $y_i = f([x_i, z_i]; w) + \varepsilon_i$. Next, we randomly permute the order of $\{z_i\}$ so that we lose the data correspondence. Here, $\mathcal{D}_1 = \{(x_i, y_i)\}$ and $\mathcal{D}_2 = \{z_j\}$ mimic two parts of data collected from two separate platforms.

We adopt a linear model $f(x; w) = x^\top w$. To evaluate model performance, we use error$= \sum_i (\widehat{y}_i - y_i)^2 / \sum_i (y_i - \bar{y})^2$, where $\widehat{y}_i$ is the predicted label, and $\bar{y}$ is the mean of $\{y_i\}$.

**Baselines**. We use Oracle, LS, Stochastic-EM as the baselines. Notice that without a proper initialization, Stochastic-EM performs well in partially permuted cases, but not in fully shuffled cases.

For better visualization, we only include this baseline in one experiment. Furthermore, we adopt two new baselines: Sliced-GW (Vayer et al., 2019) and Sinkhorn-GW (Xu et al., 2019a), which can be used to align distributions and points sets.

**Results**. We visualize the fitting error of regression models in Figure 11. We can see that ROBOT outperforms all the baselines except Oracle. Also, our model can beat the Oracle model when the dimension is low or when the noise is large.

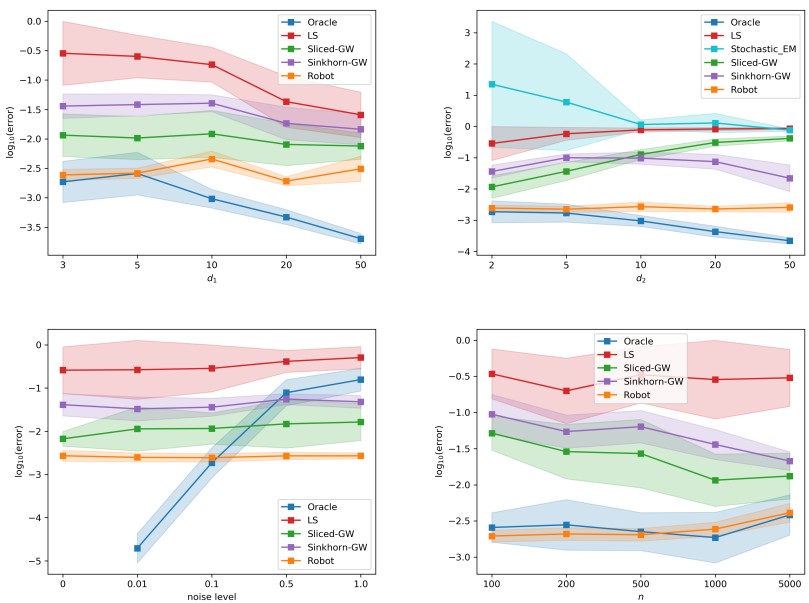

Figure 11: Linear regression. We use $n = 1000$, $d = 2$, $e = 3$, $\rho^2_{\text{noise}} = 0.1$ as defaults.

