# OpenReview forum: "A Hypergradient Approach to Robust Regression without Correspondence"
_ICLR.cc/2021/Conference — ICLR 2021 Poster_

### Official Review · AnonReviewer2 · 2020-10-28
**Very clear presentation and interesting applications**

**Rating:** 6
**Confidence:** 3

**Review:**

In this submission, the authors propose a bilevel optimization based solution to the problem of Regression Without Correspondence (RWOC).

Strong points:
1. The paper writing is very clear. I didn't know the RWOC problem before reading this submission; but after reading the introduction, I can clearly understand the problem setting, its applications (the two provided examples are great!), its challenges and what is the high-level idea of the proposed solution. Also, the organization and presentation of experiment part are very clear and easy to follow.

2. Besides the normal case of RWOC, the authors also consider and solve the case of "partial one-to-one correspondence". They also demonstrate its application using multiple-object tracking.


There are some issues that can be addressed to further make the submission strong:
1. For the experiments about multi-object tracking, could the authors include some (at least one) strong baseline? The current experiment setting for this part is not so convincing and it is a bit difficult to justify the improvement.

2. Besides the application of multi-object tracking, could the authors discuss more potential applications of RWOC? This can enlarge the application scope and make the studied problem more important and practical.


Minor places:
1. Page 2, the paragraph above "Related Works": rRWOC -> RWOC

2. Fig 2 and Fig 3: For the printed paper version (black-white), these figures are hard to read. The authors can use different markers or textual to differentiate methods.

---

> ### Author Response · Authors · 2020-11-24
> **Thanks for your feedback and we have modified the paper accordingly**
>
> Thanks for your constructive feedback.
>
> **Q1: Could the authors include some (at least one) strong baseline?**
>
> Note that our setting does not require any human annotation, while most MOT works require the annotations of the correspondence as supervision. Therefore, our setting is more difficult than the standard MOT setting, and most works for MOT are not comparable to our method. Nevertheless, we include the scores of the SORT model (Bewley et al., 2016, requiring annotation) obtained from the dataset platform, which is the only un-anonymous model at the time we wrote this paper. Our performance is comparable or better than SORT.
>
> **Q2: Could the authors discuss more potential applications of RWOC?**
>
> In the updated version, we include the following applications in Section 5 to enlarge the application scope:
>
> * SLAM tracking (Thrun, 2007) is a classical problem in robotics where the environment in which measurements are made is unknown, and part of the problem is to infer relative permutations between measurements.
>
> * In large sensor networks, it is often the case that the number of bits of information that each sensor records and transmits to the server is exceeded by the number of bits it transmits in order to identify itself to the server (Keller et al., 2009).
>
> * The pose and correspondence estimation problem in image processing (David et al, 2004).
>
> * The genome assembly problem from shotgun reads (Huang & Madan, 1999).
>
> * Archaeological measurements (Robinson, 1951).
>
> * The data correspondence is masked for privacy reasons. For example, we can build a recommender system for a new platform, borrowing user data from a mature platform.
>
> **Minor Q1: Page 2, the paragraph above "Related Works": rRWOC -> RWOC**
>
> rRWOC is short for robust RWOC, as defined in two paragraphs earlier.
>
> **Minor Q2:  Fig. 2 and Fig. 3 are hard to read**
>
> We have updated the figures in the updated version.

---

### Official Review · AnonReviewer1 · 2020-10-28
**Review --- A Hypergradient Approach to Robust Regression without Correspondence**

**Rating:** 8
**Confidence:** 5

**Review:**

The authors proposed a novel method for regression problems with outliers. The main idea is to first propose a mixed-integer optimization problem for the regression problem and then and the optimization procedure of finding the solutiuon of the problem differentiable, and the objective function of the problem are also be rephrased as a differentiable function. Based on this, an end-to-end learning approach can be established.

Pros:

1. The motivation of the paper is very clearly stated in the text, and the sketch of the theorems make the paper easy to understand.
2. The experimental part is good and it proved the efficiency of the proposed method.
3. The idea of converting a mixed-integer programming to a differentiable function is elegent.

Cons:
1. The authors says that they are going to somehow relax the one-to-one matching constraints, however, in the main text we can see that the model is still based on strict one-to-one matching constraints. In the experiments, for synthetic data, every generated data is in fact an one-to-one matching. For other datasets, through they are not strictly one-to-one, but they are close to one-to-one.

2. Theorem 1 is a trival result due to total uni-modular, and it is proved many years ago, maybe it would be better to simple give a citation there.

---

> ### Author Response · Authors · 2020-11-24
> **Thanks for your feedback and we have modified the paper accordingly**
>
> Thanks for your insightful feedback.
>
> **Q1: Experiments are exactly or close to one-to-one.**
>
> In our experiments in Section 4.2, the data size of $\mathcal{D}_1$ is only 80% of $\mathcal{D}_2$, which is not close to one-to-one. Furthermore, for experiments in Section 4.2, we adopt stochastic gradient descent, where in each iteration we sample 500 samples from $\mathcal{D}_1$ and $\mathcal{D}_2$ respectively to update $w$. More experiment details can be found in Appendix G.2.
>
> **Q2: Theorem 1 is trivial.**
>
> As we clarified in the paper, Theorem 1 is a direct corollary of the Birkhoff–von Neumann theorem. We make it as a theorem just for highlighting, to make the paper easier to understand.
> In the updated version, we change Theorem 1 to be Proposition 1, to avoid misunderstanding our contribution.

---

### Official Review · AnonReviewer3 · 2020-10-28
**Would like to see more consideration to EM, RANSAC and AD**

**Rating:** 5
**Confidence:** 4

**Review:**



The paper presents a method for robust regression with no correspondences. The association matrix is replaced by a matrix with continuous positive values. Two practical and relevant problems are investigated.

This seems to be a very interesting investigation, and the authors have done many commendable things such as studiyng relevant problems with real data, comparing to multiple alternatives, and appear to bring an interesting perspective to the problems.

The exposition of the method seems to lack a few details that might perhaps have escaped the attention of the authors. The recommendation unfortunately should tend towards not accepting. The authors should be urged to consider their work inside a wider perspective.

The main method that is repeatedly compared to the proposal was only AM. The proposal is in many ways actually more similar to Expectation-Maximization. The Birkhoff polytope here basically appears to represent likelihoods of data assignment according to Gaussian likelihoods.

It does not seem clear if the authors are exploring some further constraint in their proposal. It would be paramount to contrast the method with EM, though. The paper, as it is, looks like it could be basically proposing a form of EM without recognizing this. And for sure EM must bring many of the advantages over AM as the paper promises.

Some further smaller points:

In section one, on the second point, ending with "...they get stuck in local optima easily." What exactly is the argument here? If we are talking about a local, hill-climbing style of algorithm, there is no hope to escape local optima unless the initialization is improved, or the landscape is improved, or the algorithm is modified in such a way that it is not hill-climbing anymore, becoming some form of global optimization. In what way does the proposal differ?

EM, compared to AM, would imply in an enhanced landscape, as also seems to be the case with the proposal. Another benefit is merely to utilize slower although more robust first-order optimization methods.

It's important to make explicit what exactly are the proposed benefits. The way this sentence is written may give an impression the algorithm is proposing something that goes above a local search style of algoritm.

Another important point is about initialization which is certainly crytical to all such algoritms. In the experiments the authors suggest there's an improvement there, although the improvement must be more related only to a better landscape. Although what was the initialization afterall? This must be evaluated in the context of how good the available initializations are.

Still on section one: "Efficient first-order optimization algorithms" - Efficient relative to what? How would second-order algorithms be classified?

Regarding the experiments, RANSAC, by virtue of being a global optimization algoritm unlike the proposal and other alternatives, would be expected to be able to reach very high levels of accuracy, even if associated with a great computational cost. When comparing such a method with RANSAC one would expect to see a discussion about time/accuracy compromises. The paper is only presenting RANSAC as a method that could not reach a satisfactory accuracy. It should present at least an accuracy at a setting that was deemed to be equiparable computationally.

One final remark about AD. It is in general expected that AD techniques can match explicit derivative formulas, unless the problem is not well suited to the specific AD technique used. Or sometimes the user is required to make some extra tuning or configuration. It would be nice to review exactly how AD was not suitable, and whether there isn't an AD based solution for that (e.g. forward mode versus backwards mode).

---

> ### Author Response · Authors · 2020-11-24
> **EM, RANSAC and AD were already compared empirically, and we now add more discussions - Part 1**
>
> Thanks a lot for your detailed feedback.
>
> **Q1: Comparison to EM.**
>
> Both AM and EM are compared extensively in our experiment section (4.1 and 4.3). When presenting ROBOT, we adopt AM for comparison because AM is the motivation of ROBOT -- we adopt a hypergradient method to improve the optimization landscape.
>
> We introduce the similarity and dissimilarity between ROBOT and EM in Section 5 in the updated version. We highlight ROBOT has the following advantages comparing to EM:
>
> 1. EM requires strong underlying assumptions on the probabilistic model. For example, Abid (2018) assumes all permutations are equally likely in the prior distribution. In contrast, ROBOT is model-free.
>
> 2. EM as in Abid (2018) cannot converge to desired solutions in many cases. We now elaborate on this using the notations and the algorithm in Abid (2018, https://proceedings.allerton.csl.illinois.edu/2018/media/files/0060.pdf). First, we randomly initialize $w_0$, i.e., the estimation of $w$ at the beginning of the algorithm. Then the estimated variance $\sigma^2_0$ should be large. Substituting the variance into Eq. (10) in Abid (2018), the posterior distribution of $\Pi$ should be approximately uniform. So $\hat{\Pi}_0$ is approximately a uniform distribution. In this way, the follow-up update for $w$ in Eq. (13) is not informative -- we are basically supervising the regression model using just the mean of $y_i$. Therefore, the solution of EM would quickly converge to a stationary point with no predictive power. We illustrate one example of the converged $\hat{\Pi}$ in Figure 8. In contrast, ROBOT does not suffer from this issue.
> Note that for this reason, we initialize every EM experiment with a good initial point, found either by RANSAC or AM. More details are presented in Appendix G.
>
> 3. EM is computationally more expensive than ROBOT. Specifically, EM inevitably requires Monte Carlo steps, since the distribution of $S$ is unlikely to have a closed form expression. For example, Abid (2018) uses Metropolis-Hastings (MH) sampling, which is usually much slower than gradient descent. In contrast, ROBOT updates $w$ using a hypergradient, which only involves cheap matrix-vector multiplications.
>
>
> **Q2: "...they get stuck in local optima easily." What exactly is the argument here?**
>
> For RWOC, $w$ and $S$ can have strong interaction: When $w$ changes, $S$ can change significantly. In the AM algorithm, $S^{(k)}$ is optimized based on the suboptimal $w^{(k-1)}$, such that it can be far away from the optimal correspondence. Consequently, $w^{(k)}$ is updated by the gradient of this inaccurate $S^{(k)}$. The error caused by the “inaccurate” gradient can lead to significant suboptimality. In other words, the optimization landscape of $w$ is ill-conditioned due to the substantial interaction between $w$ and $S$, leading to non-effective updates.
>
> ROBOT, in contrast, explicitly takes into account how “inaccurate” $S^{(k)}$ is. Therefore, it has a better optimization landscape, which means it is less likely to go into local optima. Intuitively, we illustrate this effect using a quadratic example in Remark 2. Empirically, as shown in Figure 6, ROBOT can achieve smaller training loss than AM, suggesting ROBOT can find better optima than AM.
>
> **Q3: Initialization.**
>
> In all of our experiments, all algorithms are initialized similarly except EM.
>
> We initialize EM with the solution found by AM in Section 4.3, as we found that other initialization methods behave poorly. Please refer to the Appendix G for more details.
>
> **Q4: Efficient relative to what? How would second-order algorithms be classified?**
>
> First-order algorithm is in general more efficient than zero-order (derivative-free) and second-order algorithms.
>
> First-order optimization algorithms, e.g., gradient descent, are widely used to train large models, e.g., ResNet, BERT, because they have good iteration complexities and good per-iteration complexities. In comparison, zero-order algorithms have high iteration complexities, and second-order algorithms have high per-iteration complexities.

---

> > ### Author Response · Authors · 2020-11-24
> > **EM, RANSAC and AD were already compared empirically, and we now add more discussions - Part 2**
> >
> > **Q5: RANSAC**
> >
> > RANSAC (Varol & Nejatbakhsh, 2019) essentially treats the permuted data as outliers. Therefore, RANSAC is only applicable to partially permuted cases, while our ROBOT is applicable to both partially and fully permuted cases. We compared RANSAC with ROBOT in Section 4.1, where we consider the case that only 50% of the data are permuted.
> >
> > RANSAC is only able to obtain global optima without observational noise, i.e., $\varepsilon = 0$ (on Page 1). At the presence of noise, especially when the proportion of the permuted data is large, RANSAC suffers from significant suboptimality. Specifically, to fit an accurate model, RANSAC needs to sample a subset, where most of the subsampled data are un-permuted. However, when the proportion of the permuted data is large, the probability to get such data is extremely low. Therefore, we need to pay a prohibitively high computational cost to attain the global optimality.
> > When there is observational noise, i.e., $\varepsilon \neq 0$. RANSAC can only obtain approximately optimal solutions, and the corresponding computational cost is also prohibitively high.
> > In addition, the performance of RANSAC is sensitive to the inlier threshold, subsample size, and permutation proportion. Tuning these hyperparameters require significantly more effort than other methods.
> > Note that in our experiments, we have fine-tuned the hyperparameters, and subsampled for 200,000 times when using RANSAC. However, RANSAC performed worse than ROBOT and AM.
> >
> > Moreover, we remark that our ROBOT is not proposed to replace RANSAC. The strengths of RANSAC and ROBOT can be combined. For example, in Section 4.1, the initializations of AM, EM and ROBOT are the output of RANSAC. As shown in the experiment results, this further increases the performance.
> >
> > To show that combining RS and ROBOT can achieve better results than RS alone, we compare the following two cases: i). Subsample $2\times10^5$ times using RS; ii). Subsample $10^5$ times using RS followed by ROBOT for $50$ training steps. The results in Figure 10 suggest the RANSAC+ROBOT combination has better performance when the proportion of permuted data is high. More discussion can be found in Appendix G.5.
> >
> > **Q6: AD**
> >
> > The gradients computed by AD are nearly identical to ROBOT, except that AD requires much more memory and time, as suggested by Figure 7 (b)(c).
> >
> > Specifically, in the forward pass, we adopt iterative algorithms for both RWOC and rRWOC. Generally speaking, we need to run the inner algorithms for many iterations (e.g., 100 iterations) to ensure a small convergence error. If the inner iteration number is too small, the solutions of the lower problem, i.e., $S$, will not be feasible. In this way, the update for $w$ is not informative, and as a result the convergence of higher level optimization will be difficult to converge. Even if it converges, the obtained optima is not as good as using 100 inner iterations, as suggested by Figure 7 (a).
> >
> > Therefore, many iterations are needed in the forward pass. Since Sinkhorn algorithm has linear convergence, the obtained correspondence $S_t$ (where $t$ is the number of inner iterations), is close to $S^*_{\epsilon}$.
> >
> > Note that ROBOT solve for gradient by substituting $S^*_{\epsilon}$ into the expression of gradient. The gradients computed by AD or ROBOT are nearly identical.
> >
> > If using AD to compute the gradient, we need to store all the intermediate states in the inner iterations. In contrast, ROBOT computes the gradient with an explicit expression, which does not need the intermediate steps to be stored. So ROBOT is more memory efficient. In addition, using an explicit expression is faster than backpropagation through the iterations.
> >
> > This problem is particularly suitable for directly computing the gradient because the lower problem is simple, smooth and strongly convex. Therefore we can directly compute the hypergradient, and the computation of the gradient is stable.

---

### Official Review · AnonReviewer4 · 2020-10-29
**The paper present a novel method for regression which involves continuous parameters as well as finding a permutation between two sets of observations. Compared with baselines, the methods performs  well.**

**Rating:** 7
**Confidence:** 3

**Review:**

Strengths:
- the problem formulation is clean and clearly explained
- the method presentation is well written
- the techniques in used in the optimization steps, after  the relaxation step, are well motivated

Weaknesses:
- it is not clear whether the significance of  Regression without Correspondence, is high. The Multi-object tracking experiment seems contrived, at least a very good approximation of the permutation matrix can be obtained using descriptors and motion continuity
- the issue of initialization, is not fully resolved. In the reported experiment in the discussion section, ROBOT succeed 30% of the time. The percentage may vary and it is not clear how a user of the method should find a good initialization regime.
- in the experimental set-up that most resemble realistic applications (GFC for the cytometry), the improvement over AM, say a "trivial baseline" is not significant.

Note: calling a method ROBOT will make it very difficult to google

---

> ### Author Response · Authors · 2020-11-24
> **Thanks for your feedback and we have modified the paper accordingly**
>
> Thanks a lot for your valuable feedback.
>
> **Q1: The significance of RWOC.**
>
> RWOC naturally arises in many machine learning applications. The input-output correspondence might be not accessible for two reasons:
> 1.
> The measuring instruments are unable to preserve the correspondence between the samples and the measurements. In addition to gated flow  cytometry and multi-object tracking, we list a few more examples:
>
>     * SLAM tracking (Thrun, 2007),  is a classical problem in robotics where the environment in which measurements are made is unknown.
>     * Archaeological measurements (Robinson, 1951) suffer from an inherent lack of ordering, which makes inference of chronology hard.
>     * In large sensor networks, it is often the case that the number of bits of information that each sensor records and transmits to the server is exceeded by the number of bits it transmits in order to identify itself to the server (Keller et al., 2009).
>     * The pose and correspondence estimation problem in image processing (David et al., 2004).
>     * The genome assembly problem from shotgun reads (Huang & Madan, 1999).
>
> 2. The data correspondence is masked for privacy reasons. For example, when building a recommender system for a new platform, we can borrow user data from a mature platform, where the user identities are not accessible.
> We include more discussions on the applications of RWOC in Section 5 in the updated version.
>
> **Q2: Initialization.**
>
> Unfortunately, addressing the difficulty of requiring multi-start is out of the scope of this work. We are working on a follow-up work combining ROBOT with branch-and-bound algorithms (see Tsakiris, 2019 for similar ideas), where we can carefully choose the initializations to guarantee a global optimal solution.
>
> **Q3: In the experimental set-up that most resemble realistic applications (GFC for the cytometry), the improvement over AM is not significant.**
>
> In GFC, ROBOT already achieves better results than oracle. This suggests that the dataset is noisy, and AM and ROBOT are both more robust to outliers than oracle. In this case it is difficult to tell whether AM or ROBOT is better.
>
> In FC, which is also a realistic application, ROBOT performs much better than AM.

---

### Author Response · Authors · 2020-11-24
**Summary of our modifications to the paper**

We would like to thank all reviewers for their helpful feedback!

We believe that we have addressed all issues in the updated version of the paper. We highlight our modifications as follows:

1. Included more discussion on the connections and differences between EM and ROBOT, as suggested by R3.

2. Added experiments and discussions on RANSAC in Appendix G.5, as suggested by R3.

3. Added experiments and discussions on AD in Section 5, as suggested by R3.

4. Clarified the settings on initializations in Section 4.1 and Appendix G, as suggested by R3.

5. Included more discussion on the wide applicability of RWOC in Section 5, as suggested by R2 and R4.

6. Changed Theorem 1 to Proposition 1, as suggested by R1.

7. Revised the notations in the paper, making it more concise.

---

### Decision · Program_Chairs · 2021-01-07
**Final Decision**

**Decision:**

Accept (Poster)

**Comment:**

This paper proposes a method to solve regression without correspondence.  The problem is well-motivated, and the proposed method is technically sound. The motivation, organization, and presentation of the paper are very clear.  Reviewers’ suggestions to further improve the paper (e.g., clarifications on initialization, comparison and discussion with with EM, AD, etc) were adequately incorporated to the revised manuscript.